# A key function for microtubule-associated-protein 6 in activity-dependent stabilisation of actin filaments in dendritic spines

Leticia Peris [1], Mariano Bisbal [1,2], José Martinez-Hernandez [1,6], Yasmina Saoudi[1], Julie Jonckheere[1], Marta Rolland[1], Muriel Sebastien [1], Jacques Brocard[1], Eric Denarier [1,3], Christophe Bosc [1], Christophe Guerin[4,5], Sylvie Gory-Fauré[1], Jean Christophe Deloulme[1], Fabien Lanté[1], Isabelle Arnal[1], Alain Buisson[1], Yves Goldberg[1,3], Laurent Blanchoin[4,5], Christian Delphin[1] & Annie Andrieux [1,3]

Emerging evidence indicates that microtubule-associated proteins (MAPs) are implicated in synaptic function; in particular, mice deficient for MAP6 exhibit striking deficits in plasticity and cognition. How MAP6 connects to plasticity mechanisms is unclear. Here, we address the possible role of this protein in dendritic spines. We find that in MAP6-deficient cortical and hippocampal neurons, maintenance of mature spines is impaired, and can be restored by expressing a stretch of the MAP6 sequence called Mc modules. Mc modules directly bind actin filaments and mediate activity-dependent stabilisation of F-actin in dendritic spines, a key event of synaptic plasticity. In vitro, Mc modules enhance actin filament nucleation and promote the formation of stable, highly ordered filament bundles. Activity-induced phosphorylation of MAP6 likely controls its transfer to the spine cytoskeleton. These results provide a molecular explanation for the role of MAP6 in cognition, enlightening the connection between cytoskeletal dysfunction, synaptic impairment and neuropsychiatric illnesses.

[1] GIN, Inserm 1216, Univ. Grenoble Alpes, 38000 Grenoble, France. [2] Instituto de Investigación Médica Mercedes y Martin Ferreyra, INIMEC-CONICET-Universidad Nacional de Córdoba, 5016 Córdoba, Argentina. [3] CEA, Inserm 1216, BIG-GPC, Univ. Grenoble Alpes, 38000 Grenoble, France. [4] CytoMorpho Lab, UMR5168, Biosciences & Biotechnology Institute of Grenoble, CEA, CNRS, INRA, Univ. Grenoble-Alpes, 17 rue des Martyrs, 38054 Grenoble, France. [5] CytoMorpho Lab, UMRS1160, Institut Universitaire d'Hématologie, Hôpital Saint Louis, INSERM, CEA, Univ. Paris Diderot, 1 Avenue Claude Vellefaux, 75010 Paris, France. [6] Present address: Ikerbasque, Department of Biochemistry and Molecular Biology, University of the Basque Country (UPV/EHU), Basque Foundation for Science, 48940 Leioa, Spain. These authors contributed equally: Leticia Peris, Mariano Bisbal. Correspondence and requests for materials should be addressed to L.P. (email: leticia.peris@univ-grenoble-alpes.fr) or to M.B. (email: mbisbal@immf.uncor.edu) or to A.A. (email: annie.andrieux@univ-grenoble-alpes.fr)

Dendritic spines, the neuronal membrane protrusions that form the post-synaptic part of most excitatory synapses in the adult mammalian brain, can display both striking structural flexibility and remarkable persistence. These features have long been known to depend on the spine actin cytoskeleton[1,2]. Dendritic spines are rich in actin, and remodelling of spine actin networks is known to drive structural modifications associated with synaptic plasticity, such as the expansion of spine volume that accompanies activity-induced increases in synaptic efficacy[3–5]. Spines contain different pools of actin filaments, displaying distinct dynamics, with sub-membrane foci of dynamic filaments likely providing the mechanical force needed for spine expansion, whereas the much more stable polymeric actin accumulated in the spine core seems to be required for stabilising large mature "memory" spines[3]. The molecular mechanisms that drive these dynamics remain incompletely defined. In addition to actin, the role of dendritic microtubules in plasticity events was also highlighted with the discovery of their transient entrance in spines during synaptic activity, in tight correlation with an increase in actin polymerisation and spine enlargement[6–9]. Proteins associated with microtubule ends, such as EB3, interact with elements of the post-synaptic machinery, several of which are known to regulate actin dynamics[6,10]; how such interactions contribute to spine plasticity remains unclear. The question arises whether other microtubule-associated factors might control aspects of the spine actin cytoskeleton.

The neuronal MAPs (microtubule associated proteins) including MAP1B, MAP2, Tau and MAP6 are attractive candidates in this regard. Long overlooked as mono-functional proteins controlling microtubule properties, MAPs have been recently shown to interact with actin[11–16]. MAPs exist at synapses and MAP-deficient animals exhibit various synaptic defects[17–21]. In particular, MAP6 KO (also known as STOP KO) mice display severe behavioural and cognitive deficits, associated with strong impairments in both short-term and long-term synaptic plasticity[17,22]. MAP6 is known to bind and stabilise microtubules (MTs) through two types of MT-binding sequence elements: the so-called Mc modules (1 to 6 modules depending on species) that confer protection against MT disassembly at low temperature, and the Mn modules (3 modules) that convey resistance to both cold-induced and nocodazole-induced disassembly[23,24]. Importantly, it was shown that CaMKII phosphorylation of MAP6 induced its relocalization from MTs toward actin-rich domains in neurons[25]. MAP6 has been detected by immunostaining in synaptic compartments[25] and also found in several analyses of the post-synaptic proteome[26–28]. Here, we investigate the possible participation of MAP6 in the regulation of the dendritic spine cytoskeleton. Using MAP6 KO mice, we show that MAP6 is involved in the formation and the maintenance of mature post-synaptic spines, and that this novel aspect of MAP6 function relies on the Mc modules. While at physiological temperature Mc modules are unable to bind microtubules[24], we find that they can bind actin filaments. Further, we show that, in dendritic spines, Mc modules mediate actin stabilisation in response to neuronal activation, a key event of synaptic plasticity. Finally, we study the interaction of Mc modules with actin, using purified proteins in a range of in vitro assays. We show that Mc modules do not affect the actin polymerisation rate but enhance filament nucleation, protect existing filaments against depolymerisation, act on filament conformation, and organise stable filament bundles in an ordered fashion. Altogether our findings indicate that MAP6 and the specific type of actin rearrangements it promotes are needed for proper maturation and plasticity-related modifications of excitatory synapses. These results also provide a molecular explanation for the crucial role of MAP6 in cognitive abilities.

## Results

**A role for MAP6 in spine maturation and maintenance.** To visualise dendritic spines in MAP6-deficient neurons in vivo, MAP6 KO mice were cross-bred with *Thy1*-eYFP-H transgenic mice[29]. Transgenic MAP6 KO offspring animals expressed yellow fluorescent protein in layer 5 cortical neurons, as did transgenic MAP6$^{+/+}$ controls. Dendritic spine density and morphology were analysed in brain sections from these mice, using confocal microscopy (Fig. 1a). MAP6 KO neurons displayed significantly fewer dendritic spines than WT neurons (Fig. 1b; $1.18 \pm 0.04$ and $0.91 \pm 0.04$ spines/µm for WT and MAP6 KO neurons, respectively). Similar results were obtained in cultured hippocampal MAP6 KO or WT neurons (Fig. 1d), and in WT neurons transfected with MAP6-targeting siRNAs[30] or scrambled control (Fig. 1g). MAP6 KO neurons exhibited a reduced density of dendritic spines (Fig. 1e; $1.15 \pm 0.05$ and $0.92 \pm 0.05$ spines/µm, for WT and MAP6 KO neurons, respectively). MAP6 depletion by siRNA similarly resulted in a reduction of spines (Fig. 1h; $1.25 \pm 0.04$ and $0.91 \pm 0.08$ spines/µm for control and MAP6 siRNAs, respectively).

Dendritic spines are often classified in three morphological types, corresponding to successive developmental stages: thin, stubby and mushroom-like spines[31]. We quantified the density of spines subtypes, in WT and MAP6 KO neurons in vivo (Fig. 1c). While no difference was found between genotypes in the number of thin spines, mushroom spines were significantly reduced in MAP6 KO neurons as compared to WT neurons (Fig. 1c; $0.38 \pm 0.02$ and $0.26 \pm 0.02$ spines/µm for mushroom spines, in WT and MAP6 KO neurons, respectively). We performed the same quantitative analysis in cultured neurons (Fig. 1f–i). Similar to the in vivo data, the lack of MAP6 led to a decreased density of mushroom-like spines. We also observed a significant loss of stubby spines in MAP6 KO neurons in vivo (Fig. 1c; $0.31 \pm 0.02$ and $0.21 \pm 0.01$ spines/µm in WT and MAP6 KO neurons, respectively); this latter was not reproduced in cultured KO neurons, perhaps due to differences in spine maturation conditions.

To determine whether the loss of spines correlated with changes in glutamatergic synaptic transmission, we performed whole-cell patch-clamp recordings of spontaneous miniature excitatory post-synaptic currents (mEPSCs) in cultured hippocampal neurons (Fig. 1j). In KO neurons, the mean mEPSCs amplitude was not significantly changed compared to WT neurons (Fig. 1k; $25.76 \pm 0.79$ and $23.84 \pm 0.7$ pA for WT and KO neurons, respectively). In contrast, the mean mEPSCs frequency recorded in KO neurons was strongly reduced compared to WT neurons (Fig. 1l; $1.25 \pm 0.25$ and $0.52 \pm 0.17$ Hz for WT and KO neurons, respectively). The reduction of mEPSCs frequency in KO neurons corroborates the reduction of spine density and is indicative of a perturbed glutamatergic neurotransmission.

Taken together, these results indicate that MAP6 has an important role in the formation and/or maintenance of mature dendritic spines.

**MAP6 Mc modules mediate effects on dendritic spine density.** We next sought to determine which part of the MAP6 sequence was involved in spine formation and maintenance. Rescue experiments were carried out in KO neurons in vitro, using GFP-labelled MAP6 isoforms or deletion mutants (Fig. 2a) in combination with unfused mCherry as an outline marker. Spine density was measured using the mCherry image (Fig. 2b). Expression of either the MAP6-N-GFP or the MAP6-E-GFP isoform in MAP6 KO neurons resulted in an increase of spine density from $0.86 \pm 0.06$ to $1.07 \pm 0.02$ and $1.05 \pm 0.03$ spines/µm, respectively, values

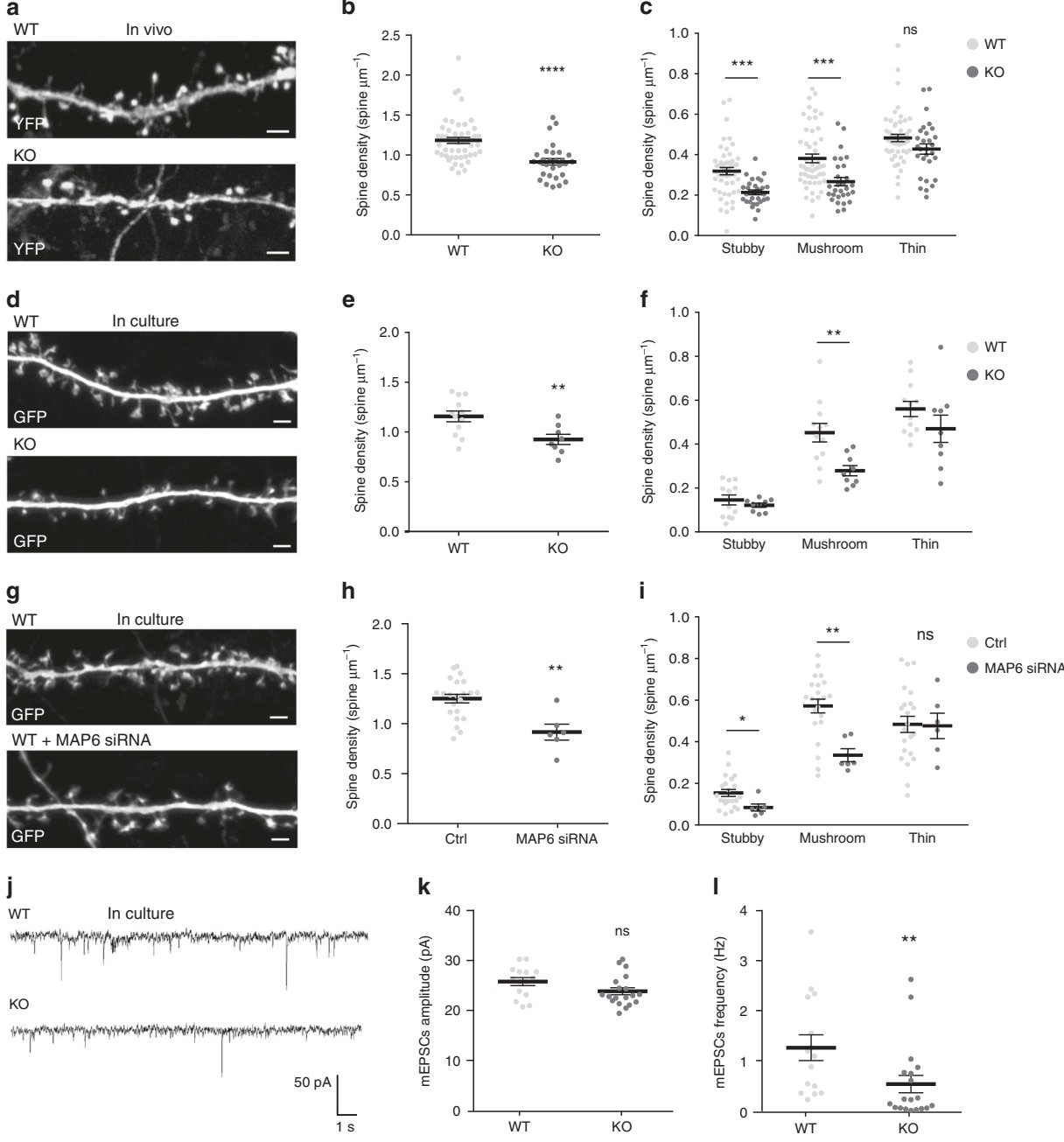

**Fig. 1** MAP6 inhibition reduces dendritic spine density. **a** Confocal image showing representative examples of dendritic segments of layer 5 cortical neurons of WT and MAP6 KO *Thy1*-eYFP-H mice. Scale bar: 2 μm. **b**, **c** Graphs of dendritic spine density in apical secondary or tertiary dendrites. The total density (**b**), or that of each morphological spine type (**c**) in WT and MAP6 KO *Thy1*-eYFP-H neurons are represented as mean ± SEM (ns: not significant, ***$p <$ 0.001; Student's *t*-test), $n =$ 50 neurons/ 4 WT animals and 29 neurons/ 4 KO animals. **d** Confocal image showing representative examples of dendritic segments of WT and MAP6 KO hippocampal neurons in culture at 18 DIV, transfected with GFP-expressing vector. Scale bar: 2 μm. **e**, **f** Graphs of total dendritic spine density (**e**) or density by shape type (**f**) in WT and MAP6 KO cultured neurons. Values are represented as mean ± SEM (ns: not significant, ***$p <$ 0.001 Student's *t*-test), $n =$ 12 neurons/ 3 WT embryos and 9 neurons/ 3 KO embryos, respectively. **g** Confocal image showing representative examples of dendritic segments of 18 DIV cultured hippocampal WT neurons transfected with GFP-expressing vector and control siRNA or MAP6 siRNA. Scale bar: 2 μm. **h**, **i** Graphs of total dendritic spine density (**h**) or density by shape type (**i**) in neurons transfected with control or MAP6 siRNA. Values are represented as mean ± SEM (ns: not significant, *$p <$ 0.05; **$p <$ 0.01; Student's *t*-test), $n =$ 22 neurons/ 4 embryos and 6 neurons/ 3 embryos from control and siRNA treated neurons, respectively. **j** Sample traces showing mEPSCs of WT and MAP6 KO 18 DIV cultured hippocampal neurons. **k**, **l** Graphs of the mean mEPSCs amplitude (**k**) and frequency (**l**) recorded in WT or MAP6 KO cultured neurons. Values are represented as mean ± SEM (ns: not significant, **$p <$ 0.01; Student's *t*-test), $n =$ 15 neurons/ 5 WT embryos and 19 neurons/ 3 KO embryos, respectively. For each experiment of cultured neurons, analysed neurons were pooled from at least three independent cultures

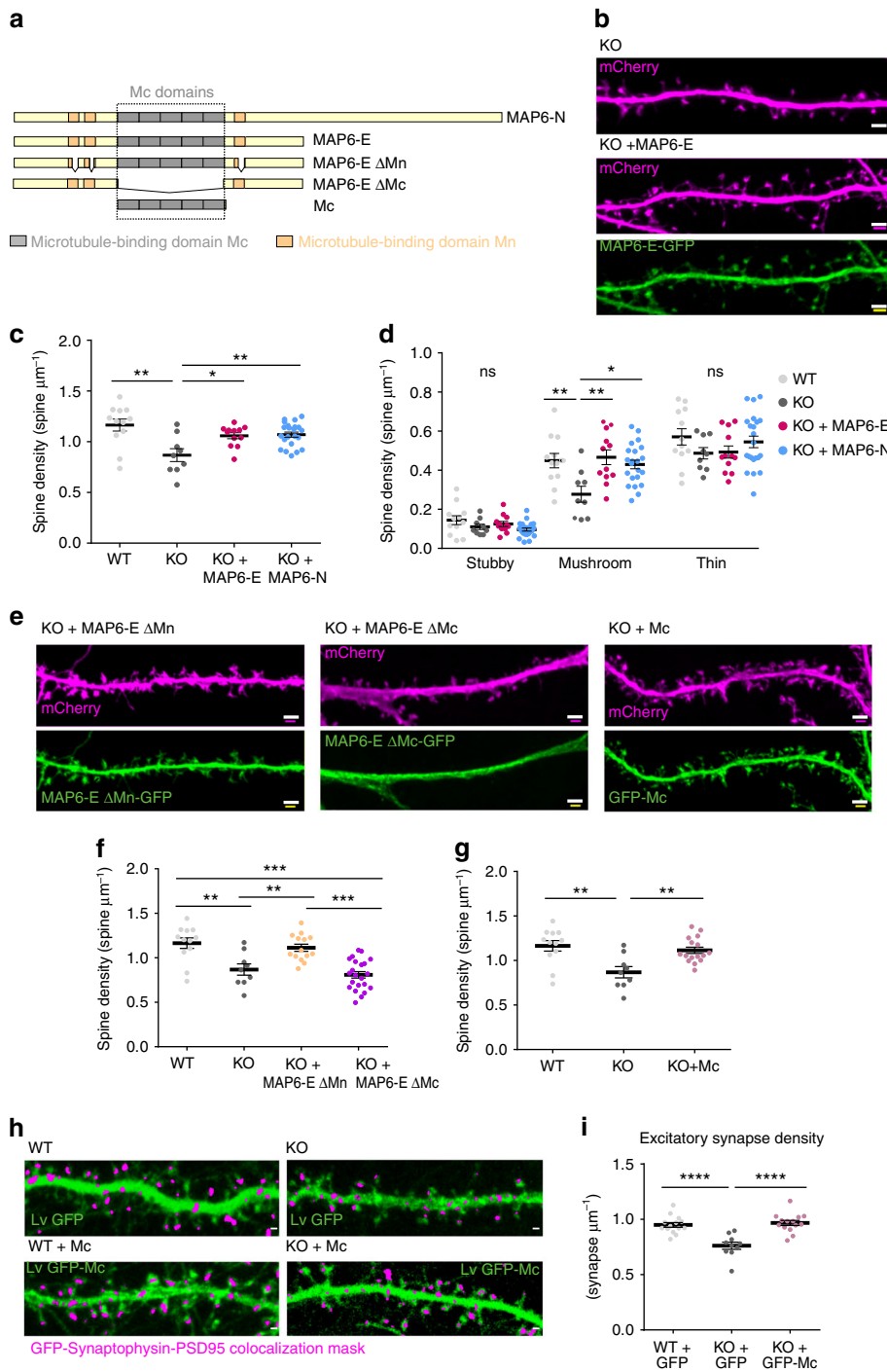

no longer significantly different from the density of 1.16 ± 0.06 spines/μm, observed for WT neurons (Fig. 2b, c). Analysis of spine morphology indicated that both MAP6-E-GFP and MAP6-N-GFP restored a normal density of mushroom-like spines in MAP6 KO neurons (Fig. 2d). We then examined the ability of MAP6-E mutants devoid of microtubule-binding sequence modules to correct spine density defects. Figure 2a depicts the two types of MAP6 microtubule-binding modules, the Mn modules (orange) and the Mc modules in the central domain (grey). When transfected in MAP6 KO neurons, MAP6-E lacking Mn modules

(MAP6-E-ΔMn-GFP) was able to increase the spine density from 0.86 ± 0.06 to 1.11 ± 0.04 spines/μm (Fig. 2e, f). In contrast, transfection of MAP6-E lacking Mc modules (MAP6-E-ΔMc-GFP) failed to significantly change the spine density (Fig. 2e, f). Conversely, expression of GFP fused to Mc modules alone (GFP-Mc) increased spine density up to values similar to those observed in WT neurons (Fig. 2e, g; 1.11 ± 0.04 and 1.16 ± 0.06 spines/μm for rescued MAP6 KO neurons and WT neurons, respectively).

To determine whether changes in morphologically defined spines corresponded to synaptic alterations, we next used

 

**Fig. 2** MAP6 Mc modules are important for dendritic spine maturation and maintenance. **a** Schematic representation of MAP6 domain structure and cDNA constructs used in this paper. **b** Confocal image showing representative examples of dendritic segments of MAP6 KO 18 DIV cultured hippocampal neurons transfected with mCherry-expressing vector alone or co-transfected with a MAP6-E construct. Scale bar: 2 μm. **c**, **d** Graphs showing the quantification of dendritic spine density in WT neurons, MAP6 KO neurons and MAP6 KO neurons transfected with MAP6-E or MAP6-N constructs. Total spine density (**c**) or type-wise density (**d**) are represented as mean ± SEM (*$p < 0.05$; **$p < 0.01$; one-way ANOVA and Tukey's post hoc test), $n = 12$ neurons/ 3 embryos, 9 neurons/ 3 embryos, 12 neurons/ 4 embryos and 22 neurons/ 5 embryos for WT, KO, KO + MAP6-E and KO + MAP6-N, respectively. **e**–**g** Rescue experiments in MAP6 KO neurons (**e**) Confocal image showing representative examples of dendritic segments of MAP6 KO 18 DIV cultured hippocampal neurons transfected with mCherry-expressing vector and MAP6-E ΔMn-eGFP, MAP6-E ΔMc-eGFP or eGFP-Mc domain constructs. Scale bar: 2 μm. **f**, **g** Graphs of dendritic spine density in WT neurons, MAP6 KO neurons and after rescue of MAP6 KO neurons by MAP6-E ΔMn or MAP6-E ΔMc (**f**) or Mc modules (**g**). Data presented as mean ± SEM (**$p < 0.01$, ***$p < 0.001$; one-way ANOVA and Tukey's post hoc test), $n = 12$ neurons/ 3 embryos, 9 neurons/ 3 embryos , 14 neurons/ 4 embryos and 22 neurons/ 4 embryos neurons for WT, KO, KO + MAP6-E ΔMn and KO + MAP6-E ΔMc, respectively. **h** Representative images of dendrites of WT and MAP6 KO 18 DIV hippocampal neurons transduced with GFP or GFP-Mc lentivirus (green) and superposed with a magenta mask corresponding to pixels that simultaneously contain PSD-95 (post-synaptic), Synaptophysin (pre-synaptic) and GFP label. Scale bar, 2 μm. **i** Graph showing the quantification of excitatory synapse density in dendrites of WT and MAP6 KO neurons transduced with GFP or GFP-Mc lentivirus. Data presented as mean ± SEM. (****$p < 0.0001$ one-way ANOVA with Tukey's post hoc test), $n = 14$ neurons/ 3 embryos, 10 neurons/ 3 embryos and 14 neurons/ 3 embryos  for WT + GFP, KO + GFP and KO + GFP-Mc, respectively. For each experiment, neurons were pooled from at least three independent cultures

immunological markers to label excitatory synapses[32] in cultured neurons. Neurons were transduced with lentiviral vectors expressing GFP or Mc-GFP, fixed at DIV18, and stained for PSD-95/synaptophysin (a post-synaptic and a pre-synaptic marker, respectively). Fluorescent puncta containing both pre and post-synaptic markers were used to detect and count synapses formed on the dendrites of transduced cells, as shown in Fig. 2h. GFP-expressing WT neurons displayed 0.949 ± 0.020 synapses/μm (Fig. 2i), whereas GFP-expressing MAP6 KO neurons had 0.76 ± 0.030 synapses/μm (Fig. 2i), a value compatible with the presence of one synapse per dendritic spine. Thus, in MAP6 KO neurons, the reduced density of excitatory synapses is in tight correlation with the reduction of spine density. Expression of Mc modules (GFP-Mc) in MAP6 KO neurons rescued the number of synapses (Fig. 2h, i, 0.967 ± 0.022 synapses/μm) to values similar to those observed in WT neurons.

Altogether, these results demonstrate that Mc modules contribute to the growth and maintenance of dendritic spines and to the establishment of mature synapses.

**A single Mc module ensures normal dendritic spine density**. Human MAP6 only contains one Mc module, whereas Mc modules are present in five copies in rat MAP6[33] (R1 to R5, Supplementary Fig. 1A). We showed that rat R5, human MAP6-E or the human Mc module were able to correct the spine density deficit (Supplementary Fig. 1C-F). Thus, a single Mc module, as present in human MAP6, is sufficient to ensure dendritic spine maturation and maintenance.

**Phosphorylation by CaMKII targets MAP6 to spines**. To determine whether the effect of MAP6 on synapses requires it to enter spines, we took advantage of the physiological regulation of MAP6 targeting by CamKII[25]. We showed that MAP6-E phosphorylation mutant (S/E mutant, Supplementary Fig. 1A, G–H) mimicking phosphorylated state was able to rescue spine density whereas MAP6-E phosphorylation mutant preventing phosphorylation (S/A mutant, Supplementary Fig. 1A, G-H) was not efficient. These results indicate that MAP6 transfer into the spines through phosphorylation by CamKII correlates with the effect of MAP6 on synapse maintenance.

**MAP6 Mc modules directly interact with actin**. At physiological temperature, MAP6 interaction with microtubules relies on Mn domains (Fig. 2a), and Mc modules are not involved[23,24]. We

thus wondered whether an interaction of the Mc modules with actin might occur, possibly accounting for the observed spine stabilisation. We analysed the comparative distribution of endogenous MAP6 and cytoskeletal elements in cultured hippocampal neurons at different stages of maturation. Endogenous MAP6 co-localised with microtubules throughout the cell, was especially enriched in axons, but also co-localised with actin in filopodia and lamellipodia within the growth cone (Fig. 3a, a′, a″). After 30 days in culture, endogenous MAP6 continued to be present in cell body, axon and dendrites but also accumulated within dendritic spines (Fig. 3b, b′, b″). To determine whether Mc modules could target the protein to actin-rich regions of the cell, we studied the localisation of a transiently expressed GFP-Mc fusion protein. The GFP-Mc polypeptide was found to co-localise with actin filaments both in mouse embryo fibroblasts (Supplementary Fig. 2) and in the growth cone of young neurons (Fig. 3c, c′, c″).

To determine whether MAP6 directly interacted with actin filaments in neurons, we performed FRET (Förster Resonance Energy Transfer with Acceptor Bleaching) measurements using the acceptor photobleaching technique[34], with TRITC-phalloidin labelled actin filaments (as acceptor) and GFP or GFP-tagged MAP6 (as donor) (FRET-A; Fig. 3d, d′; Supplementary movie 1). We chose to analyse growth cones, where actin filaments are clearly visible. The fluorescence intensity of the donor MAP6-E-GFP within growth cone regions increased significantly after bleaching of the acceptor, revealing a FRET efficiency of 52 ± 2% (Fig. 3e). Similar experiments were performed with GFP-Mc modules, yielding a FRET efficiency of 49.6 ± 2% (Fig. 3e). As a control (Fig. 3d, d′), no transfer of fluorescence was observed between F-actin filaments and soluble GFP (1.4 ± 3% of FRET, Fig. 3e). The FRET-AB technique did not allow us to obtain accurate measurements of FRET efficiency in dendritic spines.

Overall, these results show that endogenous MAP6 appears to be present very extensively throughout the neuron, including at actin-rich structures, where, as indicated by the acceptor photobleaching experiment, it can be in close proximity to F-actin.

**MAP6 regulates actin dynamics in dendritic spines**. The above results raise the possibility that MAP6 might participate in the regulation of actin dynamics in spines, which is crucially involved in spine remodelling and in synaptic plasticity events[35]. In particular, long-term potentiation (LTP) is known to be accompanied by a rise of actin filaments, increased actin stabilisation and spine enlargement[36]. To examine whether MAP6 contributes

 

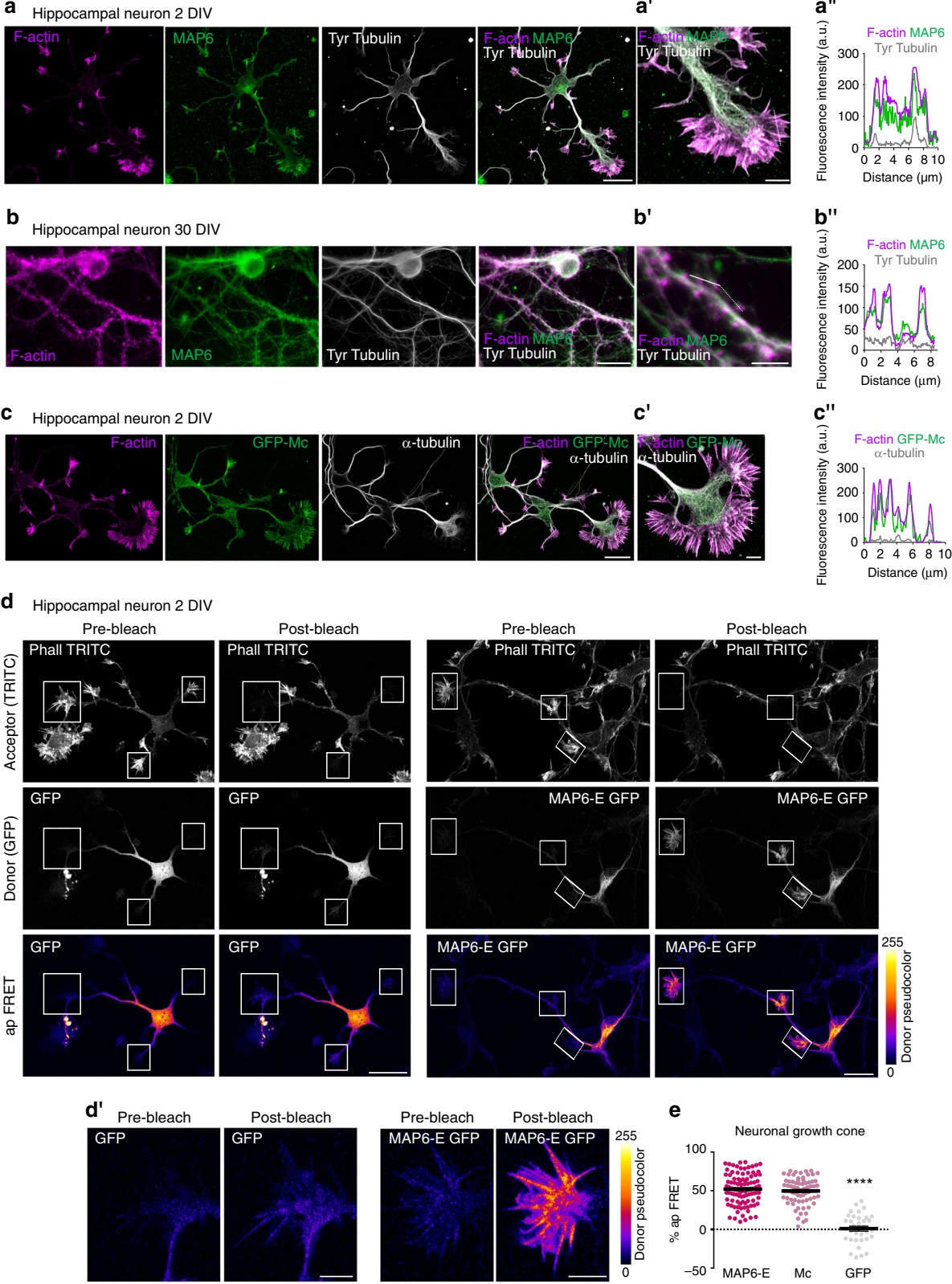

to plasticity-related actin filament accumulation and spine expansion, we used neurons transfected with plasmids encoding GFP as a volume marker, together with the filamentous actin probe LifeAct-mCherry (Fig. 4a). The transfected neurons were subjected to synaptic activation, using a chemical protocol

mimicking LTP (cLTP, bicuculline and 4-Aminopyridine)[21]. Individual mushroom-like spines of live neurons were imaged before and after induction of cLTP (Fig. 4a). Consistent with previous publications[4,37–39], in WT neurons, cLTP resulted in a strong expansion of spine volume (Fig. 4b). By contrast, no

**Fig. 3** The MAP6 Mc module directly interacts with the actin cytoskeleton. **a** Confocal image showing a representative cultured hippocampal neuron at 2 DIV, co-stained for actin filaments (phalloidin, magenta), endogenous MAP6 (green) and microtubules (grey). Scale bar: 20 μm. **a'** Higher magnification view of the axonal growth cone. Scale bar: 5 μm. **a"** Graph of the fluorescence intensity variations of actin filaments, MAP6 and microtubule staining along a line scan (white dotted line). **b** Confocal image showing a 30 DIV cultured hippocampal neuron stained as in **a**. Scale bar: 20 μm. **b'** Higher magnification view of a dendrite. Scale bar: 5 μm. **b"** Graph of the fluorescence intensity variations of actin filaments, endogenous MAP6 and microtubule staining along a line scan. **c** Confocal image showing a 2 DIV cultured hippocampal neuron transfected with GFP-Mc construct (green) and stained for actin filaments (magenta) and microtubule (grey). **c'** Higher magnification view of the axonal growth cone. Scale bar: 5 μm. **c"** Graph of the fluorescence intensity variations of actin filaments, GFP-Mc and microtubule staining along a line scan. **d** Acceptor Photobleaching FRET (ap FRET) analysis of 2 DIV cultured hippocampal neuron transfected with either GFP or MAP6-E GFP construct and stained for actin filaments (TRITC-phalloidin). Images were taken before (pre-) and after (post-) acceptor photobleaching for the acceptor (TRITC-phalloidin, upper images) and the donor (GFP or MAP6-E GFP, middle images). For better visualisation of the donor fluorescence intensity, a fire lookup table (LUT) was applied to pre- and post-bleach images, with colours corresponding to pixels from 0 (black-violet) to max intensity (yellow). The insert boxes show bleach regions of interest. Scale bar: 20 μm. **d'** Higher magnification view of the axonal growth cone before and after acceptor photobleaching. Scale bar: 5 μm. **e** Graphs showing the Acceptor Photobleaching FRET efficiency in neuronal growth cones of hippocampal neurons transfected with MAP6-E-GFP, GFP-Mc or GFP constructs. Data presented as mean ± SEM (****$p < 0.0001$; one-way ANOVA and Tukey's post hoc test), $n = 97, 72$ and 35 axonal growth cones from MAP6-E, Mc and GFP-expressing neurons of 8, 7 and 4 embryos respectively, from three independent cultures

increase was detected in the spines of KO neurons (Fig. 4b). Further, in agreement with the results of Bosch et al.[4], in WT spines, the concentration of actin filaments (estimated as the ratio of LifeAct vs. GFP) was ~20% higher following cLTP (Fig. 4c). Strikingly, no such change in actin concentration was detected in KO spines (Fig. 4c). Thus, MAP6 is required for the growth of the actin cytoskeleton that underlies spine expansion during activity-induced synaptic change. To determine whether MAP6 could regulate the polymerisation of spine actin, we compared the turnover rate of actin monomers in spines of MAP6 KO neurons and WT controls, by using fluorescence recovery after photobleaching (FRAP) of GFP-actin. Under our conditions, the mobile fraction of GFP-actin molecules calculated from the recovery kinetics corresponded to actin treadmilling in dynamic filaments, while the immobile fraction was composed of stable filaments[35]. Hippocampal neurons cultured for 15 days in vitro were transfected with GFP-actin (Fig. 4d), 2 days later a specific spine was bleached and fluorescence recovery measured over time (Fig. 4d). In WT neurons, FRAP measurements yielded a classical average recovery curve with a characteristic recovery time of 40 s and almost full recovery of fluorescence (Fig. 4e, WT). Similar results were obtained for MAP6 KO neurons (Fig. 4e, KO) indicating similar actin dynamics in spines at resting state. Following cLTP, in WT neurons, FRAP measurements showed slower recovery of the mobile fraction and appearance of a significant immobile fraction, indicating as expected a diminished rate of actin turnover and an increased pool of stable actin filaments (Fig. 4e, WT + cLTP). In contrast, in MAP6 KO neurons, the cLTP protocol did not affect the recovery curve (Fig. 4e, KO + cLTP). Quantification, in WT neurons, of the dynamic GFP-actin in resting conditions or after cLTP clearly showed a reduction of the mobile fraction from $92.8 \pm 3\%$ to $57 \pm 9\%$ (Fig. 4f, 43% of immobile actin), whereas the mobile fraction was unchanged in MAP6 KO neurons (Fig. 4f, $95 \pm 4\%$ and $100.0 \pm 5\%$, in resting and activated conditions, respectively). These results indicate that the actin stabilisation triggered by chemical LTP requires MAP6.

We then determined the ability of full-size or truncated MAP6 to restore the actin stabilisation defects observed in spines (Fig. 4g, h). In neurons transfected with MAP6-E, cLTP-induced actin stabilisation was restored, as cLTP brought the mobile fraction down to $66.8 \pm 4\%$, very similar to $57\% \pm 9\%$ obtained in WT conditions (Fig. 4h) and significantly different from the $100 \pm 5\%$ found in MAP6 KO neurons (Fig. 4h). MAP6-E-ΔMc was unable to allow actin stabilisation (Fig. 4g, h), whereas Mc was able to restore actin stabilisation, lowering the mobile fraction to $67.4 \pm 4\%$ (Fig. 4h). These results clearly

indicate that Mc modules stabilise actin in dendritic spines during plasticity events.

**MAP6 Mc modules promote assembly of ordered actin bundles.** We first assayed the in vitro binding of Mc modules to actin filaments using high-speed co-sedimentation assays. A significant fraction of Mc modules was found to associate with the actin pellet (Fig. 5a). Titration experiments showed that binding was saturable, with a $K_d$ of $670 \pm 106$ nM and a $B_{max}$ of $280 \pm 13$ nM for 500 nM actin (Fig. 5b).

To analyse in detail the interaction of Mc modules with actin filaments, we assayed in vitro the effect of purified Mc on a range of actin polymerisation parameters, including nucleation, elongation and bundling, known to be regulated by actin-binding proteins in spines[1,5,36]. Actin nucleation was first assayed by following the polymerisation of pyrene-labelled actin over time. As shown in Fig. 5c, the initial lag corresponding to actin nucleation was shortened when actin was polymerised in the presence of Mc modules. Next, we investigated actin assembly by directly visualising actin polymerisation using total internal reflection fluorescence (TIRF) microscopy (Fig. 5d). After a 5 min assembly period, we measured the number of actin filaments present in the microscope field of view, in the presence or absence of Mc. Quantification showed that the total filament number was increased in the presence of Mc modules (Fig. 5e; total filament number: $66 \pm 9.9$ and $121.6 \pm 15.5$; for actin and actin + Mc modules, respectively). However, the elongation rate of polymerising filaments was not significantly changed by Mc ($1.1 \pm 0.02$ and $1.1 \pm 0.02$ μm/min for actin and actin + Mc, respectively). In contrast, we observed a dose-dependent induction of actin bundling by Mc modules, as indicated by the higher density of filaments, visualised by the orange to yellow pseudo-colour in Fig. 5f (fluorescence intensity ≥ 200 arbitrary units). Quantification indicated a higher percentage of actin bundling in the presence of Mc modules (Fig. 5g; $5.7 \pm 0.6\%$, $14.2 \pm 1.5\%$ and $19 \pm 2.5\%$ for 300 nM of actin alone, +150 and +300 nM Mc, respectively). To monitor the kinetics of Mc-induced actin bundle formation, we performed light-scattering assays during the polymerisation of actin monomers with or without Mc (Fig. 5h). In the presence of Mc, light scattering increased much more rapidly, indicating that Mc quickly bound and bundled newly forming actin filaments. Furthermore, the ability of Mc to bundle actin filaments was examined in a co-sedimentation assay (Fig. 5i). Complete actin polymerisation was checked by high-speed (HS) sedimentation ($100,000 \times g$, actin filaments in the pellet). At low-speed (LS) centrifugation ($15,000 \times g$), in the control condition, actin filaments were mainly present in the supernatant whereas,

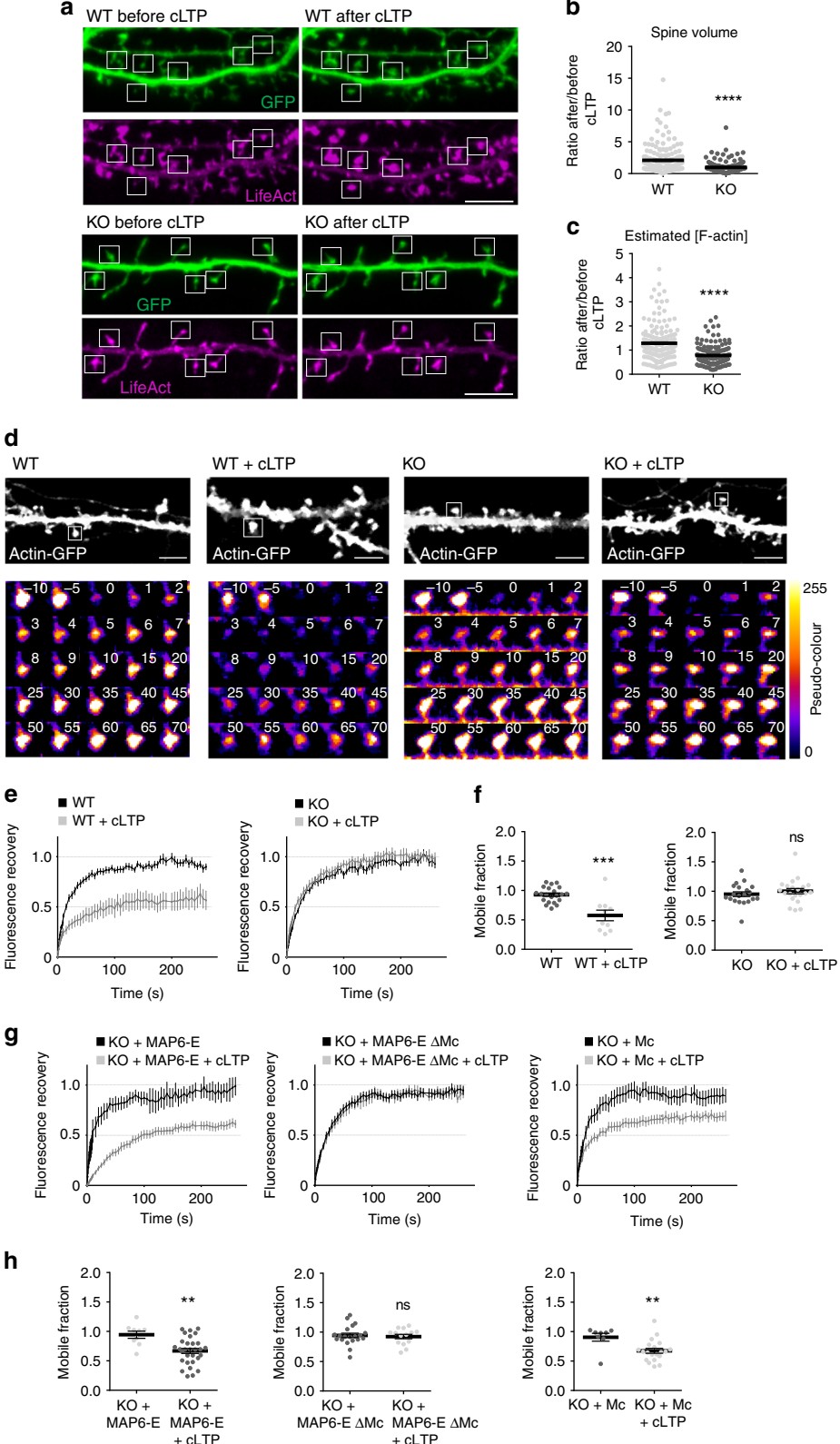

in the presence of Mc, 4-time increase of actin was found in the pellet (13 ± 4.9% and 58.5 ± 3.9% for actin alone and actin + Mc in the LS pellet). Finally, we analysed the impact of Mc modules on actin organisation using negative-staining electron microscopy. As shown Fig. 5j, in the presence of Mc modules, actin filaments formed bundles and appeared straightened as compared

to control conditions. Moreover Mc modules caused filaments to organise in single layered arrays, showing periodic striations spaced by 35.9 ± 1.2 nm (Fig. 5j, white arrowheads). These striations could also be seen in single filaments (spaced by 35.7 ± 1.1 nm) and were not observed in the absence of Mc. The period coincides with the helical pitch of actin (35.7 nm),

**Fig. 4** MAP6 regulates the actin filaments stable pool in activated dendritic spines. **a** Representative confocal images of dendritic spines (boxes) of live cultured WT and KO hippocampal neurons at 18 DIV, co-transfected with GFP and LifeAct-RFP, before and after cLTP. Scale bar: 5 μm. **b** Ratio of post- to pre-cLTP spine volumes. Data presented as mean ± SEM. (***$p < 0.001$; Mann–Whitney test). $n = 172$ and 169 spines from 10 WT and 10 KO neurons before and after cLTP, respectively. **c** Ratio of post- to pre-cLTP spine F-actin concentration. Data presented as mean ± SEM. (***$p < 0.001$; Mann–Whitney test). $n = 186$ spines/ 10 neurons/ 4 WT embryos and 185 spines/ 10 neurons/ 7 KO embryos before and after cLTP, respectively. **d** Representative examples of neurons at 16 DIV, transfected with actin-GFP before and after cLTP. A single spine (boxed) was photo-bleached and the time course of fluorescence recovery determined. Time series of fluorescence levels are shown in pseudo-colour for better visualisation, with colours corresponding to pixels from 0 (black-violet) to max intensity (yellow). Scale bar: 5 μm. **e** Time course of actin-GFP fluorescence recovery in activated and non-activated neurons, averaged over experiments. Data presented as mean + SEM for each time point. **f** Mean value of mobile fractions derived by fitting individual curves to an exponential recovery model. Data presented as mean ± SEM (ns: not significant, ***$p < 0.001$; Student's $t$-test), $n = 21$ spines/ 7 neurons/ 7 embryos, 10 spines/ 4 neurons/ 4 embryos, 20 spines/ 5 neurons/ 5 embryos and 21 spines/ 5 neurons/ 5 embryos from WT, WT + cLTP, KO and KO + cLTP neurons. **g** Time course of actin-GFP fluorescence recovery in activated and non-activated neurons transfected with actin-GFP and MAP6-E, MAP6-E ΔMc or Mc modules, averaged over experiments. Data presented as mean + SEM for each time point. **h** Mean value of mobile fractions calculated as in **f**. Data presented as mean ± SEM (ns: not significant, **$p < 0.01$, Student's $t$-test), $n = 9$ spines/ 5 neurons/ 5 embryos, 30 spines/ 9 neurons/ 9 embryos, 20 spines/ 5 neurons/ 5 embryos, 14 spines/ 3 neurons/ 3 embryos, 8 spines/ 3 neurons/ 3 embryos and 20/ 4 neurons/ 4 embryos from KO + MAP6-E, KO + MAP6-E + cLTP, KO + MAP6-E Δ Mc, KO + MAP6-E Δ Mc + cLTP, KO + Mc and KO + Mc + cLTP neurons. For each experiment, neurons were pooled from two to four independent cultures

suggesting repeated association of Mc with a specific aspect of the filaments.

Altogether these in vitro results indicate that Mc modules regulate actin assembly and favour the formation of actin bundles.

**MAP6 Mc modules stabilise actin filaments**. To test whether Mc modules can stabilise actin filaments, we performed a dilution experiment. Filaments were polymerised from 2 μM monomeric actin, next incubated alone or with 0.5 to 1 μM of Mc modules and then the actin concentration was lowered by dilution to a final concentration of 0.1 μM. After 20 min, the amount of remaining filaments was analysed using high-speed co-sedimentation assay, and compared to the initial amount of filaments (input) (Fig. 6a). As previously described[40], dilution of actin filaments led to depolymerisation (Fig. 6b) with only 38.61 + 2.43% of filaments resisting dissociation. In the presence of Mc modules, actin filaments were more resistant to depolymerisation (Fig. 6b; 52.3 ± 1.6% and 54.7 ± 1.8% for 0.5 and 1 μM of Mc, respectively). To track depolymerisation over time, we performed similar dilution experiments with pyrene-labelled actin (Fig. 6c). Actin filaments showed a rapid decrease after buffer dilution (−25% of the initial fluorescence) whereas Mc-decorated actin filaments were more resistant (−11% of the initial fluorescence; Fig. 6c).

Taken together, these results indicate that Mc modules may stabilise actin filaments by preventing their depolymerisation.

**Human Mc module bundles and stabilises actin filaments**. Interestingly, we also showed that the hMc module was able to bundle actin filaments (Supplementary Fig. 3A–B) with periodic striations spaced by 36.02 ± 1.138 nm and to protect actin filaments from depolymerisation (Supplementary Fig. 3C–D).

## Discussion

It is well established that the maturation and plasticity of dendritic spines are strongly dependent on the complex, dynamic remodelling of the spine actin cytoskeleton. We show here that part of this remodelling is mediated by MAP6, a protein known to be indispensable for synaptic plasticity and for the normal development of brain circuits underlying aspects of cognition and behaviour. MAP6 KO and heterozygous mice exhibit cognitive defects, associated with strong impairments affecting both short and long-term synaptic plasticity, including defective LTP and LTD[17,41]. Moreover, MAP6 deficits have been linked to autism[42] in line with the involvement of spine pathology in human

neuropsychiatric disease[43]. MAP6 is known as a microtubule-associated protein that can also interact with actin depending on its phosphorylation state[25]. In this work, using biochemical and microscopy approaches, we identify MAP6 as a novel regulator of actin filament dynamics and conformation, with implications for the mechanisms of synaptic homoeostasis and plasticity.

MAP6 KO neurons, both in vivo as in vitro, displayed impairments in the morphological and functional maturation of post-synaptic compartments. The deficiency phenotype could be rescued by expressing different MAP6 isoforms. Remarkably, within the MAP6 sequence, the Mc modules were both necessary and sufficient to rescue dendritic spine density. This result cannot readily be explained by the known affinity of Mc modules for microtubules, as microtubule binding cannot occur at physiological temperature; instead, we find that Mc modules interact with actin filaments in a way consistent with direct regulation of the spine cytoskeleton.

Several results strengthen the idea that MAP6 binding to spine actin is instrumental to spine maintenance. In addition to microtubular localisation, full-size MAP6 as well as the isolated Mc domain are present in actin-rich regions of neurons, such as growth cones of young neurons or dendritic spines in mature cells. FRET assays supported the notion that, via the Mc modules, MAP6 closely interacts with actin filaments in situ. Using MAP6 phosphorylation mutants, localisation of the protein in the spine was shown to correlate with spine restoration. Functionally, FRAP measurements of spine actin dynamics revealed that MAP6 had a necessary role in the kinetic stabilisation of actin filaments that accompanies activity-induced synaptic change. Rescue experiments again showed that MAP6 depended on its Mc modules for regulating spine actin turnover; furthermore, Mc modules expressed on their own were able to restore actin stabilisation following cLTP in MAP KO neurons.

On the whole, these results indicate that MAP6 has an important part in the stabilisation of mature spine types ("mushroom-like" and "stubby" spines), i.e. those that transmit larger synaptic currents and encode long-lasting information[44]. As the fraction of thin spines was not affected, MAP6 deficiency does not seem to result in a simple reversal of spine maturation. One explanation could be that the cytoskeletal structure promoted by MAP6 binding to actin is specifically required for maintaining mature spines.

The in vitro reconstitution of the Mc-actin interaction uncovered molecular properties that are consistent with the specific function of MAP6 in spine plasticity. We found, in our in vitro conditions, that purified Mc modules interact with actin filament with a $K_d$ below the micromolar range and promote the formation of straight, stable, rigid actin bundles. Biophysical

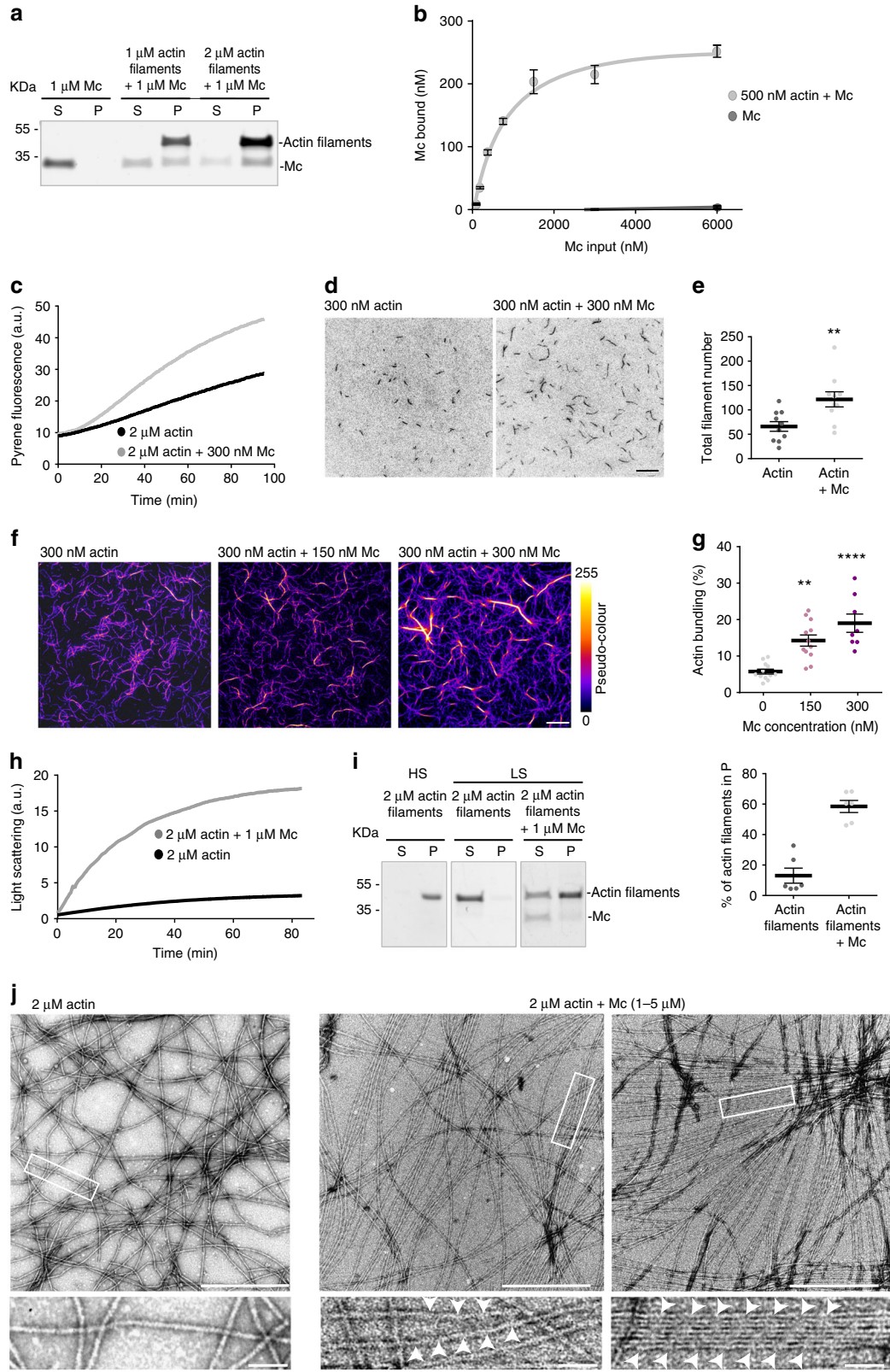

modelling indicates that morphogenesis of the spine head can be quantitatively accounted for by a balance of forces, involving a combination of actin bundles pushing outward along the axis of the spine neck, and filaments that emerge from Arp2/3-nucleated branches and that exert isotropic radial pressure in the head[45].

Indeed, spines have been shown to harbour both a sub-membrane pool of actively polymerising filaments and a core population of stable actin filaments; the increase and stabilisation of core actin is a feature of LTP[3]. The action of MAP6 Mc modules on actin filament nucleation might be important during the initial

**Fig. 5** Mc modules modify actin filament dynamics and organisation in vitro. **a** Mc modules co-sedimented without or with filamentous actin, analysed by SDS-PAGE and Coomassie blue staining (P: pellet, S: supernatant). **b** Concentration of Mc polypeptide bound to actin filaments (500 nM), plotted as a function of the total amount of Mc input. Data points represent mean ± SEM, $n = 3$ for Mc alone and $n = 4$ for Mc + actin. **c** Representative graph showing the time course of polymerisation of pyrene-labelled actin (10%) in the absence or presence of Mc modules. **d** TIRF microscopy images of actin polymerised with or without Mc modules for 5 min. Scale bar: 10 μm. **e** Quantification of the total filament number, obtained from **d** images. Data presented as mean ± SEM. (**$p < 0.01$; Student's $t$-test). $n = 10$ independent experiments. **f** TIRF microscopy images of actin filaments polymerised in the absence or presence of Mc modules for 60 min. For better visualisation of actin bundles, a pseudo-colour lookup table (LUT) was applied. Fire LUT pixel intensity map show individual actin filaments in violet and actin bundles in red to yellow. Scale bar: 10 μm. **g** Quantification of the percentage of bundled actin filaments related to total actin filaments network as a function of Mc modules concentration. Data presented as mean ± SEM. (**$p < 0.01$, ****$p < 0.0001$; Kruskall–Wallis with Dunn's multiple comparison test). $n = 12$, 12 and 8 independent experiments for actin, actin + 150 nM Mc and actin + 300 nM Mc, respectively. **h** Representative time course of actin bundling in the absence or presence of Mc modules, monitored by light scattering at 400 nm (arbitrary units). **i** Actin filaments in high (100,000×g) and low-speed (15,000×g) co-sedimentation assays in the absence or presence of Mc modules (P: pellet, S: supernatant). Data presented as mean ± SEM. (****$p < 0.0001$; Student's $t$-test). $n = 6$ independent experiments. **j** Actin alone or in the presence of Mc modules visualised by negative staining and electron microscopy. Scale bar: 500 nm. Lower panels are higher magnifications of the insert in upper panels. Scale bar: 50 nm

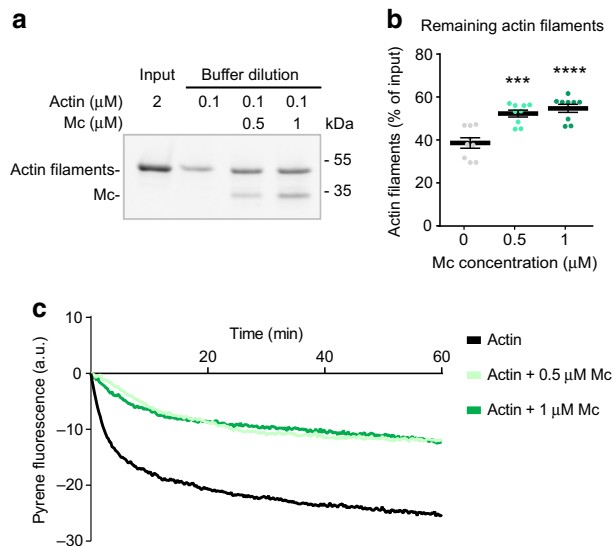

**Fig. 6** Mc modules protect actin filaments from depolymerisation. **a** Actin depolymerisation induced by dilution of actin filaments alone or in the presence of Mc modules. After 20 min of incubation in a buffer containing 0.1 μM actin (alone or with 0.5 μm or 1 μM Mc), samples were centrifuged at 100,000×g and pellets separated on SDS/PAGE and detected by Stain-Free system. **b** Quantification of the percentage of remaining actin filaments after buffer dilution. Data presented as mean ± SEM. (***$p < 0.001$, ****$p < 0.0001$, one-way ANOVA and Sidak's post hoc test), $n = 9$ gels from three independent experiments. **c** Mean time course of actin filaments and Mc-decorated actin filaments depolymerisation. After 1 h incubation of 40% pyrene-labelled actin filaments alone or in the presence of Mc modules, all samples were diluted to a final concentration of 0.1 μM actin (alone or with 0.5 μM or 1 μM of Mc) and pyrene fluorescence (arbitrary unit) was measured over time

reorganisation of the actin cytoskeleton, whereas Mc-induced filament straightening and bundling could be necessary for the enlargement and maintenance of potentiated spines (mushroom-like forms).

Several other proteins, such as drebrin, spinophilin/neurabin, or CaMKIIβ, have been described as crucial for dendritic spine morphogenesis in association with their ability to bundle actin[46–50]. However, the in vivo phenotype of MAP6 KO spines indicates that MAP6-induced actin arrangements carry specific properties, which are not conferred by other actin-binding proteins. The fact that the same MAP6 Mc module carries both nucleation-enhancing and bundling activities might allow MAP6 to promote the enlargement of the spine actin pool in a specific,

uniquely efficient fashion. Furthermore, at the ultrastructural level, both single and bundled actin filaments were seen to be strikingly straightened in the presence of the MAP6 Mc modules. The straightened fibres displayed regularly spaced striations, with a period of 35–36 nm, potentially corresponding to the helical pitch of bare actin filaments. This suggests that filaments cross-linked by Mc modules were aligned without torsion and in a highly ordered, extended conformation. The periodic striations seemingly correspond to a specific feature of MAP6-decorated actin filaments. Other MAPs such as tau and MAP2C, or the key neuronal actin-binding protein drebrin, were shown to induce assembly of actin filaments into rope-like structures or loose bundles, with no visible striations[12,14,51,52]. Within the large group of actin-binding proteins, several members, including fimbrin, coronin, FRG-1 and p135[ABP], induce actin bundling without striations[53–55]. In contrast, periodic striations similar to those seen here have been observed for fascin[56] and cortexillin[57] or septins[58]. However, these latter proteins are either absent from dendritic spines or confined to their base and neck. Thus, MAP6 is the first MAP and the first protein present within dendritic spines able to induce actin bundling with periodic striations. One limitation of the present study is that the effect of Mc MAP6 on actin organisation could only be resolved in vitro, with isolated components, and we did not further test for interaction of Mc with other similar charge and mass; future work will need to further confirm the specificity of this interaction, and to address the challenge of imaging Mc-dependent actin ultrastructure in dendritic spines.

The extended conformation of Mc-bound filaments suggests that they are under tension. Such tension has been shown to decrease binding of the filament-severing protein, cofilin[59]; hence MAP6 may conceivably protect a pool of spine actin against cofilin activity. Straight filaments induced by MAP6 might also increase the efficacy of myosin-mediated vesicular transport of lipid membrane and AMPA receptors into spines[60]. Indeed, alteration of myosin isoforms resulted in impaired AMPAR trafficking, and, similar to MAP6 deficiency, in decreased frequency of miniature EPSCs and defective activity-induced synaptic change.

Under physiological conditions, the two main MAP6 isoforms in neurons, MAP6-N and MAP6-E, contain Mc modules but also Mn modules, which have strong microtubule binding affinity. Release of MAP6 from the dendritic microtubules is necessary for translocation into spines, and might be triggered by phosphorylation events occurring upon neuronal activation. We have previously shown that MAP6 was phosphorylated by CamKII, the major kinase in spines[25]. Phosphorylation by CamKII-rendered MAP6 unable to associate with microtubules, while leaving actin-

binding unchanged (no change of binding following alkaline phosphate treatment)[25]. Here, we correlated the translocation of MAP6 phosphorylation mutants from microtubules in the dendritic shaft to actin filaments in dendritic spines, with the ability of these mutants to restore spine density. The simplest explanation of our data is that the phosphorylation-defective MAP6 mutant (S to A) was retained on MTs and thus unable to translocate into spines and thereby to rescue the spine deficit in MAP6 KO neurons. Thus phosphorylation of MAP6 may mediate some of the effects of CamKII on structural plasticity.

Finally, beside its direct effects on actin filament dynamics and organisation, we cannot exclude that MAP6 also indirectly control spine actin via signalling events, such as activation of small GTPases. Indeed, MAP6 has been shown to directly interact with Rac1[61], a GTPase known to induce strong cytoskeletal rearrangements. When present in dendritic spines and bound to actin, MAP6 might also regulate plasticity pathways by interacting with SH3 domain-containing proteins such as Intersectin or PI3Kinase, through active PRD domains as reported in the case of Sema3E-dependant axonal guidance[30].

In summary, the present data provide evidence that MAP6 acts as a novel player in the regulation of actin cytoskeleton structure and dynamics, with important consequences for the maturation and maintenance of dendritic spines in hippocampal neurons. Reduced expression of MAP6 affects the density of dendritic spines and mature synapses in vitro and in vivo, and prevents the stabilisation of the spine actin cytoskeleton during plasticity events. The ability of MAP6 to regulate actin dynamics relies on its Mc modules, which we show are able to enhance actin filament nucleation, promote filament stabilisation and bundling, and induce the appearance of a specific, straightened filament conformation. Thus, these data shed new light on the mechanisms that might contribute to the cognitive impairments associated with MAP6 deficiency.

## Methods

**Animals**. All experiments involving animals were conducted in accordance with the policy of the Institut des Neurosciences de Grenoble (GIN) and in compliance with the French legislation and European Union Directive of 22 September 2010 (2010/63/UE). The research involving animals was authorised by the Direction Départementale de la protection des populations - Préfecture de l'Isère - France and by the ethics committee of GIN n° 004 accredited by the French Ministry of Research. The homogeneous inbred 129SvPas/C57BL6, 129SvPas/C57Bl6-Thy1-eYFP-H WT and MAP6 KO mice were generated as previously described[62].

**Plasmids and recombinant proteins**. PCR amplification and cloning of cDNAs were performed with Phusion DNA polymerase (Thermo Scientific) and In-Fusion HD Cloning kit (Clontech), respectively. All constructs were verified by sequencing (Eurofins). A cDNA encoding human actin (GenBank accession number NM_001101) was PCR-amplified and inserted into the pEGFP-C1 vector. The plasmid encoding mCherry was from Clontech (pmCherry vector) and the actin filament marker LifeAct-RFP plasmid described in ref.[21] A cDNA encoding rat MAP6-N (accession number NM_017204) was PCR-amplified and inserted into pEGFP-N1. Rat MAP6-N fragments aa 1–614 (MAP6-E) and aa 221–455 (Mc), rat MAP6-E deletions of aa 124-138 (replaced by Ala) + aa 162–171 + aa 481–495 (MAP6-E-ΔMn) and aa 225–450 (MAP6-E-ΔMc), rat MAP6-E point mutations S139A + S198A + T484 A + S537A (MAP6-E-4A) and S139E + S198E + T484E + S537E (MAP6-E-4E), according to the numbering of rat MAP6-N (NP_058900), were created by PCR and cloned into pEGFP-N1 (except into pEGFP-C1 for Mc). Rat MAP6-N fragments aa 359–451 (R4R5) and aa 404–451 (R5) according to the numbering of rat MAP6-N (NP_058900) was created by PCR and cloned into pEGFP-C1. DsRed2 cDNA was amplified by PCR from pDsRed2-C1 plasmid (Clontech) and inserted in place of GFP in the GFP fusions of the above described rat MAP6 fragments MAP6-E, Mc and MAP6-E ΔMn. A cDNA encoding human MAP6-E (accession number AB527183) was PCR-amplified and inserted into pEGFP-C1. Human MAP6-E fragment aa 224–280 (hMc), according to the numbering of human MAP6-E (AB527183) was created by PCR and cloned into pEGFP-N1. The lentiviral plasmid encoding eGFP was from Addgene (pWPXLd vector, kind gift from D. Trono). A cDNA encoding rat MAP6-N fragment aa 221–235 (5R), according to the numbering of rat MAP6-N (NP_058900), was created by PCR and cloned into pWPXLd.

**Lentivirus production**. Lentiviral particles encoding either GFP-Mc or unfused GFP were produced in our institute by co-transfection of the pWPXLd-based vector with the psPAX2 and pCMV-VSV-G helper plasmids (Addgene, Cambridge, MA) into HEK293T cells obtained from ATCC (ATCC-CRL-3216). Viral particles were collected by ultra-speed centrifugation.

**Protein purification**. The Mc domain was produced in insect cells and purified as already described[24]. Human Mc synthetic peptide (hMc) was purchased from ThermoFisher Scientific, resuspended in BRB buffer (80 mM PIPES pH 6.75, 1 mM EGTA, 1 mM MgCl$_2$), aliquoted, frozen in liquid nitrogen and stored at −80 °C.

**Cell culture, transfection and immunofluorescence**. Hippocampi (E18.5) were digested in 0.25% trypsin in Hanks' balanced salt solution (HBSS, Invitrogen, France) at 37 °C for 15 min. After manual dissociation, cells were plated at a concentration of 5000–15,000 cells/cm$^2$ on poly-L-lysine-coated (1 mg/ml poly-L-lysine hydrobromide, Sigma Aldrich) coverslips for fixed samples, or on ibidi glass bottom 60 µDishes for live imaging. Neurons were incubated 2 h in DMEM-10% horse serum and then changed to MACS neuro medium (Miltenyl Biotec) with B-27 supplement (Invitrogen, France). Hippocampal neurons were incubated for 2, 16, 18 or 30 days in vitro (DIV) at 37 °C, 5% CO$_2$ in a humidified incubator. The neurons were transiently transfected with the cDNAs described above using Lipofectamine 3000 (Invitrogen) according to the manufacturer's instructions, and analysed 24 h later. For silencing experiments, efficient knockdown of MAP6 was obtained using previously described[30] Stealth RNAi siRNA1 and siRNA2 (Invitrogen). For immunocytochemistry, cells were fixed with 4% paraformaldehyde in 4% sucrose-containing PBS for 20 min and permeabilised with 0.2% Triton X-100/PBS for 5 min. Fixed cells were then incubated with primary antibodies for 3 h in 0.1% PBS/Tween and then with fluorophore-conjugated secondary antibodies for 1 h at room temperature. Primary antibodies were: rabbit polyclonal anti GFP (ThermoFisher Scientific. Cat# A-11122) diluted 1:1000, mouse monoclonal anti PSD-95 (clone K28/43, NeuroMab UC Davis/NIH NeuroMab Facility. Item # 75-028) diluted 1:500, rabbit polyclonal anti Synaptophysin (SYP (H-93), Santa Cruz Biotechnology, Inc. Cat# sc-9116) diluted 1:500, mouse monoclonal anti mCherry (Clontech Laboratories, Inc. Cat# 632543) diluted 1:1000, rabbit polyclonal antibody against MAP6 (Ab23N) diluted 1:250 and mouse monoclonal antibody against MAP6-N[63] (Ab 175) diluted 1:250, rat monoclonal anti Tyr-tubulin (YL1/2) diluted 1:1000 and mouse monoclonal antibody against alpha-tubulin[64] (clone α3A1) diluted 1:1000. Secondary antibodies were coupled to Alexa-488, to Cy3 or to Cy5 (Jackson Immuno-Research Laboratories). TRITC-Phalloidin (Sigma Aldrich, Inc. Cat# P1951) was used to label actin filaments. Fluorescent images were acquired with a Zeiss LSM 710 confocal microscope using a ×40 and ×63 oil-immersion objective (NA 1.25 and NA 1.4) and ZEN 2010 software (Carl Zeiss MicroImaging).

**Imaging of dendritic spines**. For fixed samples, images of dendritic segments of Thy1-eYFP-H mouse cortical neurons (serial sections of cortex) or 18 DIV transfected neurons visualised by soluble GFP or soluble mCherry fluorescence were obtained using a confocal laser scanning microscope (Zeiss, LSM 710). Serial optical sections (1024 × 1024 pixels) with pixel dimensions of 0.083 × 0.083 µm were collected at 200 nm intervals, using a ×63 oil-immersion objective (NA 1.4). The confocal stacks were then deconvolved with AutoDeblur. Dendritic spine analysis (spine counting and shape classification) was performed on the deconvolved stacks using Neuronstudio[65,66]. All spine measurements were performed in 3D from the z-stacks. The linear density was calculated by dividing the total number of spines present on assayed dendritic segments by the total length of the segments. At least three dendritic regions of interest were analysed per cell from at least three independent cultures in each experimental condition.

For live imaging during chemical LTP, 17 DIV WT or MAP6 KO hippocampal neurons were co-transfected with GFP and LifeAct-RFP cDNAs using Lipofectamine 3000 and imaged 24 h later at 37 °C on a heated stage with 5% CO2 (PeCon), using a confocal laser scanning microscope (Zeiss, LSM 710) and GaAsP detector (Zeiss Airyscan). Dendritic segments of neurons that simultaneously expressed GFP and LifeAct-RFP were selected to image ~30 spines per neuron at baseline state or 15 min after cLTP (50 µM bicuculline and 2.5 mM4-Aminopyridine). Only modest bleaching was observed during the experiment. Cells showing signs of damage (pearling or blebbing) were discarded. Individual spines, identified using Neuronstudio[65,67], which were identifiable at initial and final time points were manually selected. Each dendritic spine volume before and after cLTP was quantified on the GFP stack using Neuronstudio and a ratio was calculated as follows: final volume (fi)/initial volume (iv) with iv as the volume before stimulation and fv as the volume 15 min after cLTP induction. To estimate F-actin concentration in dendritic spine, a region of interest (ROI) including the whole spine was placed at each previously identified dendritic spine on LifeAct-RFP and GFP stacks at baseline state or 15 min after cLTP induction. The fluorescence intensity (FI) of the Maximun Intensity z projection of the LifeAct-RFP and GFP stacks were measured for each spine, before and after cLTP. Individual spine F-actin concentration was estimated as follows: [FI LifeAct/FI GFP] after cLTP / [FI LifeAct/FI GFP] before cLTP.

**Imaging of excitatory synapses**. For immunodetection of excitatory synapses in transduced neurons, 1/100 of a hippocampal cell suspension was infected by 15 min incubation with GFP or GFP-Mc lentivirus (Lv) at a multiplicity of infection of 40. The infected population was then mixed with non-transduced cells and plated on poly-L-lysine-coated coverslips and incubated at 37 °C, 5% CO$_2$ in a humidified incubator. Neurons were fixed at 18 DIV and immunolabeled with GFP, PSD-95 and Synaptophysin antibodies. Fluorescent images were acquired with a Zeiss LSM 710 confocal microscope using a ×63 oil-immersion objective (NA 1.4) and ZEN 2010 software (Carl Zeiss MicroImaging). Images were enhanced for small structures with the LoG3D ImageJ plugin[68] using a 2 pixel radius, and thresholded to create a mask for: GFP (transfection and contour marker), Synaptophysin (presynaptic compartment) and PSD-95 (post-synaptic compartment). The mask corresponding to the overlap of the three markers was superposed to the GFP image. We manually counted the synaptic puncta present in dendritic spines or in the shaft and calculated their linear density. Mask creation and counting were done blind to the transduction condition.

**Electrophysiological recordings in neuronal culture**. Hippocampal neurons in culture were visualised in a chamber on an upright microscope with transmitted illumination and continuously perfused at 2 ml/min with oxygenated Artificial Cerebro-Spinal Fluid (ACSF in mM: 119 NaCl, 2.5 KCl, 1.25 NaH$_2$PO$_4$, 1.3 MgSO$_4$, 2.5 CaCl$_2$, 26 NaHCO$_3$, 0.001 tetrodotoxin, 0.05 bicuculline methiodide and 11 Glucose) at room temperature. Miniature excitatory post-synaptic currents (mEPSC) were recorded at a membrane potential of −60 mV with borosilicate glass pipettes of 4–5 MΩ resistance filled for whole-cell recordings (in mM: 117.5 CsMeSO$_4$, 15.5 CsCl, 10 TEACl, 8 NaCl, 10 HEPES, 0.25 EGTA, 4 MgATP, 0.3 NaGTP, pH 7.3). Signals were amplified with an EPC 10 Amplifier (HEKA Elektronik Dr. Schulze GmbH, Wiesenstrasse, Germany). Recordings were filtered at 1 kHz and sampled at 10 kHz using the Patchmaster Multi-channel data acquisition software (HEKA Elektronik Dr. Schulze GmbH, Wiesenstrasse, Germany).

**Fluorescence resonance energy transfer-acceptor bleaching**. Fluorescence resonance energy transfer with acceptor bleaching (FRET-AB)[34] was performed on 2 DIV or 18 DIV neurons transfected with GFP-tagged constructs (FRET donor) and stained for actin with TRITC-phalloidin (FRET acceptor). For photobleaching of the acceptor, ROIs corresponding to growth cones were exposed to five scans using the 561 nm laser line at 100% power on the LSM 710 confocal microscope. The whole neuron was imaged at lower power intensity with both the 488 and 561 nm lasers before and after the bleaching of acceptor. FRET efficiency was calculated as the relative increase of the fluorescence intensity of the donor after selective photobleaching of the acceptor, according to the formula Efficiency = $(F_{Donor\ post} - F_{Donor\ pre}) \times 100/F_{Donor\ post}$. Control measurements were performed with the same imaging protocol in neurons that had not been stained with TRITC-phalloidin (donor alone), in non-transfected neurons labelled with TRITC-phalloidin-(acceptor alone), and by omitting the photobleaching step in transfected and labelled neurons.

**Fluorescence recovery after photobleaching**. FRAP experiments were performed in 16 DIV mouse hippocampal cultures transfected with GFP-actin cDNA using Lipofectamin 3000 (Lifetechnologies), as per manufacturer's instructions. FRAP was performed 24 h after transfection on GFP-actin expressing neurons at baseline state or 15 min after cLTP treatment. Images were acquired with an inverted Nikon Eclipse Ti C2 confocal microscope with a Nikon ×60 water objective with 1.33 numerical aperture. Only mature mushroom type spines were used for the experiments. A ROI including the whole spine was imaged 10 times before bleaching. GFP-actin was bleached with maximal laser power with the 405 nm laser line (five iterations, total bleach time of 0.66 s, fluorescence reduction of ~90%). The fluorescence recovery was measured by scanning the ROI with 488 nm laser light using the following protocol: 10 images at 1 frame/s before bleaching, 10 images at 1 frame/s after bleaching followed by 60 images at 1 frame/5 s for 300 s.

Fluorescent signal analysis was performed with the Nikon software Nis. All the post-bleach values were divided by the values from the non bleached area of the cells and normalised to the first 10 pre-bleach values. The first post-bleach measurement was set to zero.

Raw fluorescence values were normalised by dividing by the values at each time point by the average value before bleaching. The analysis of the FRAP recovery data was performed with GraphPad Prism. For each FRAP assay, the plateau value ($y_0$), mobile fraction ($A$), first-order rate constant ($k$) and characteristic time ($k^{-1}$) were calculated by non-linear regression, fitting the data to a mono-exponential recovery model ($y = y_0 - A^*\exp(-k^*x)$). Goodness of fit was not improved when using a two-phase recovery model, therefore we retained the simpler monophasic curve. Mean plateau values were calculated from these individual fits. We checked the different groups of FRAP data for normality (Shapiro–Wilkes test) and variance homogeneity (Fligner test) and analysed them by one-way ANOVA, followed by post hoc pairwise $t$-tests. Diffusional entry of monomeric actin was not included in the model because in our hands it occurs with a time scale that is at least ≈40-fold faster than polymerisation; any effect of diffusion is thus included in the value of the first recovery time point.

**Pyrene actin polymerisation**. 2 μM of 10% pyrene-labelled G-actin (actin monomers) was polymerised alone or in the presence of 300 nM of Mc modules, in buffer AP (Actin Polymerisation) containing 2 mM Tris,10 mM imidazole pH 7.0, 0.2 mM ATP, 0.5 mM DTT, 0.1 mM CaCl$_2$, 50 mM KCl, 1 mM MgCl$_2$, 1 mM EGTA.

**TIRF microscopy**. Perfusion chambers were prepared with functionalised silane-PEG (Creative PEGwork) glass slides, as described previously[11]. For actin polymerisation assays, the flow cell was incubated with PLL-g-PEG (2 kDa, 0.1 mg/ml in 10 mM Hepes, pH 7.4, Jenkem) and washed with 1% BSA in buffer AP. Actin growth was initiated by flowing 300 nM G-actin (containing 30% Alexa-488 labelled G-actin) alone or with 150–300 nM Mc modules in AP buffer containing 4 mM DTT, 1% BSA, 1 mg/ml glucose, 70 μg/ml catalase, 580 μg/ml glucose oxidase and 0.3% methylcellulose. Samples were visualised on an inverted microscope (Eclipse Ti, Nikon) equipped with an Ilas[2] TIRF system (Roper Scientific), a cooled charge-coupled device camera (EMCCD Evolve 512, Photometrics), a warm stage controller (LINKAM MC60), and controlled by MetaMorph software (version 7.7.5, Molecular Devices). Samples were excited with 491 nm laser light and time-lapse imaging (at 488 nm) was performed at 26 °C for actin polymerisation, during 45 min at 1 frame per 5 s with a 100-ms exposure time.

The elongation rate of actin filaments polymerisation with or without Mc modules was determined on kymographs using ImageJ software and a home-made plugin (KymoTool). Elongation of single actin filaments was derived from actin length measurements on image stacks. To analyse actin nucleation, 300 nM actin (labelled as above), alone or together with 300 nM Mc modules was incubated in AP buffer containing 4 mM DTT, 1% BSA, 1 mg/ml glucose, 70 μg/ml catalase, 580 μg/ml glucose oxidase and 0.3% methylcellulose in the perfusion chamber for 5 min at 26 °C and imaged. TIRF images were skeletonised and the number of filaments and the length of total actin network were measured. To analyse bundling activity, 300 nM actin alone or with 150 or 300 nM Mc modules was treated as before and images were taken after 45 min. TIRF images of actin filaments were enhanced using Feature detector[69] module of Icy[70]. Actin network and bundles were segmented with Fiji[71] using Otsu or Yen thresholding algorithms, respectively. The thresholded images were skeletonised and the total length of actin network or bundles was measured. The percentage of actin bundling was calculated by dividing the length of bundled actin by the total length of filamentous actin in the same field.

**Light-scattering assay to measure bundling activity**. A kinetic light-scattering assay was performed to determine the ability of Mc modules to form actin bundles during polymerisation. Light scattering by unlabelled 2 μM G-actin was monitored at 400 nm at a scattering angle of 90°. The change of light scattering was recorded after addition of 1 μM Mc modules.

**High-speed and low-speed co-sedimentation assays**. High- and low-speed co-sedimentation assays were used to examine actin polymerisation and actin bundling, respectively[11]. All proteins were pre-clarified at 140,000×$g$ before each experiment. 2 μM G-actin (actin monomers) was polymerised for 1 h at room temperature (RT) in AP (Actin Polymerisation) buffer containing 2 μM Phalloidin. To confirm the complete polymerisation of actin, samples were centrifuged for 15 min at 100,000×$g$ (high-speed co-sedimentation assay: HS).

To analyse the binding of Mc modules to actin filaments, various concentrations of polymerised actin were incubated with Mc modules in AP buffer for 1 h at room temperature. After HS sedimentation, Mc modules and actin present in the pellet or in the supernatant were analysed by SDS-PAGE and Coomassie blue staining (Supplementary Fig. 4). For titration analysis, co-sedimented Mc domains were quantified by western immunoblotting using anti-MAP6 antibody 23C[63] and chemiluminescent detection (Chemidoc$^{TM}$MP Imaging System, BioRad). Absolute protein amounts were determined by calibration with an Mc-domain concentration range, using ImageJ for image analysis. $K_d$ and $B_{max}$ values and their confidence interval were calculated with R by non-linear fitting of the data to the single-site binding isotherm, $B/B_{max} = (T - B)/(K_d + T - B)$, where $T$ is the total (input) Mc concentration, $B$ is the actin-bound Mc concentration, and $B_{max}$ is the saturating concentration.

To analyse bundling activity, 1 μM Mc modules was added (or not) in the mixture after actin polymerisation and samples were centrifuged for 15 min at 15,000×$g$ (low-speed co-sedimentation assay: LS). All supernatants and pellets were resolved by SDS-PAGE and detected by Stain-Free technology in the gel (Mini-PROTEAN® TGX Stain-Free™, Biorad) (Supplementary Fig. 4). Protein bands were quantified from triplicate blots of independent experiments using ImageJ software (National Institutes of Health, Bethesda, MD).

**Transmission electron microscopy**. To analyse the effect of Mc modules on actin organisation, we performed negative staining electron microscopy in two sets of samples: in actin + Mc modules co-polymerisation and adding Mc modules after complete actin polymerisation. In co-polymerisation assays: 2 μM G-actin was incubated with 2 μM Phalloidin and 1 to 5 μM Mc modules in AP (Actin Polymerisation) buffer for 1 h at room temperature (RT).

In post-polymerisation assays: 2 μM G-actin was incubated with 2 μM Phalloidin in AP buffer for 1 h at RT. Then, Mc modules (1–5 μM) was added and incubated for 1 h at RT.

For negative staining, 1 μl of protein solution was loaded onto a carbon-mica interface. The carbon layer was floated on 2% uranyl acetate, recovered with a 400-mesh copper grid (Agar Scientific), air dried, and observed with a JEOL 1200EX transmission electron microscope at 80 kV. Images were acquired with a digital camera (Veleta, Olympus) at ×100,000 magnification. Experiments were performed at the Electron Microscopy Facility of the Grenoble Institute of Neuroscience. To detect the periodic striations of Mc-decorated single and bundled filaments, a line was traced between the edges of each filament (outside or inside the bundle), staining intensity along the line was plotted, and the distances between successive intensity minima were measured. Each value corresponds to the mean distance between successive striations of the same filament. Images of independent experiments were analysed.

**Actin depolymerisation by buffer dilution**. To analyse actin depolymerisation, we first incubated 2 μM G-actin with 2 μM Phalloidin in AP buffer for 1 h at RT. Then, 0.5 to 1 μM of Mc modules was added and incubated for 1 h at RT. G-actin was diluted from 2 to 0.1 μM in AP buffer (alone or with 0.5 or 1 μM of Mc modules) for 20 min, centrifuged at 100,000×g and pellets were separated on SDS-PAGE and detected by Stain-Free technology in the gel (Mini-PROTEAN® TGX Stain-Free™, Biorad). Protein bands were quantified from triplicate blots of independent experiments using ImageJ software (National Institutes of Health, Bethesda, MD). The remaining actin filaments were calculated as a percentage of the input. Kinetic depolymerisation of 2 μM of 40% pyrene-labelled actin, alone or in the presence of 0.5 or 1 μM of Mc modules, was monitored after dilution to 0.1 μM actin in AP buffer (alone or with 0.5 or 1 μM of Mc modules).

**Statistical analysis**. All data are presented as mean ± SEM. Statistical significance of differences between conditions was calculated with Prism 5.0 (GraphPad Software), using tests as indicated in each figure.

**Data availability**. The datasets generated and/or analysed during the current study are available from the corresponding authors on request.

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

## Acknowledgements
We are grateful to zootechnicians of the Grenoble Institute of Neurosciences, S. Andrieu, M. Lapierre, F. Mehr and F. Rimet for animal care. We thank N. Collomb, C. Corrao, F. Vossier and C. Paoli for technical assistance; T. Rush and M. Seggio for introducing L. P. to FRAP, TIRF microscopy; Ju Brocard and G. Falivelli for helping with virus production and Fondation Bettencourt Schueller for equipment. This work was supported by INSERM, CEA, Université Grenoble Alpes and in part by awards from the French Agence Nationale de la Recherche to A.A. (2010- Blanc-120201 CBioS); to A.B., A.A., I.A. (2011-MALZ-001-0217 MALAAD) and to A.A., I.A. (2017-CE11-0026 MAMAs). This work was supported by the Photonic Imaging Center of Grenoble Institute Neuroscience (Univ Grenoble Alpes – Inserm U1216) which is part of the ISdV core facility and certified by the IBiSA label. PhD stipends were from the French Ministry for Research and Higher Education for M.R., from AFM-Téléthon for M.S. and from Région Auvergne Rhones Alpes for J.J. M.B. was a recipient of the Roche Pharmaceutic RPF program (AA team and F Hoffmann-La Roche Ltd, Basel, Switzerland).

## Author contributions
L.P., M.B. and A.A. conceived and designed the study. L.P. and M.B. performed the experiments, with contributions from J.M.-H., Y.S., J.J., M.S., J.B., E.D., C.B., S.G.-F., J.C.D. and C.D. C.G., I.A. and L.B. contributed to cell-free studies. F.L., M.R. and A.B. performed electrophysiological experiments. L.P., M.B., Y.G. and A.A. wrote the manuscript, with contribution from all co-authors.

## Additional information

**Competing interests:** The authors declare no competing interests.

