## [Peer Review File · Nature Communications]

Editorial Note: An image has been redacted as indicated to protect copyright claims. The figure can be viewed in the original publication, listed in the place of redaction.

Reviewers' comments:

Reviewer #1 (Remarks to the Author):

Peris et al demonstrate that the canonically microtubule-associated protein MAP6 co-localizes with actin in dendritic spines and the growth cone, and is there responsible for promoting nucleation and bundling of actin filaments. They report that MAP6 supports the formation or maintenance and the maturation of dendritic spines. The authors show that this effect is mediated by the Mc domain of MAP6, and relies on the CaMKII-dependent phosphorylation of MAP6 for disassociation from microtubules and subsequent actin relocalization.

Overall, the manuscript is well written, and the experiments are well thought out. The decision to focus briefly on LTP-associated changes using cLTP and the resulting figure are particularly striking. Additionally, the discussion was well-phrased and integrates previous observations in a compelling fashion, including the link to cofilin, septins, and non-MAP proteins known to cause striations in actin in other regions of the neuron. The detail in Figure 5H was stunning.

However, some clarifications and a few additional experiments would further increase the impact of the work, and minor revisions to the text would clarify the scope of this project.

Specific points to address:

1. Experimentally, there is some concern about inconsistencies with the number of neurons examined for each condition, even within a single experiment. There are a number of times that 8-9 neurons are used for one experimental condition (generally KD), while the WT control and rescue conditions have double to triple the number of neuronal repeats. These experiments would be stronger if the groups being compared included similar numbers of observations. For example, Fig 4 there are only 10 neurons for the WT+cLTP when all other conditions have at least 20 neurons, and for 3E two conditions have 8 and 9 neurons, while some other conditions have up to 30 neurons, over three times the lowest value.
2. The authors quantify spine density in many figures, but there seems to be a striking change in spine area as well. Has this been examined?
3. The largest problem is the inconsistency with which MAP6's interaction with microtubules is explained. Multiple times a separate paper is referenced which posits that MAP6 only binds microtubules at lower temperatures, yet throughout this manuscript MAP6 is seen co-localizing with microtubules. Within the manuscript it is both stated that "as Mc domains cannot bind MTs at physiologic temperature" (page 9) but also that "endogenous MAP6 co-localized with MTs" (page 10). Please demonstrate that other domains are responsible for MAP6 binding to MTs or find a different way to explain these divergent findings.

4. Throughout the manuscript, MAP6 is stated as localizing with actin in particular, but this overlooks the obvious co-localization within the dendrite and the axon with MTs. MAP6 expression appears to be pan-neuronal. It would help greatly if staining was able to be conducted for phospho-MAP6 to demonstrate spine and growth cone specificity, or if KD neurons had forced expression of tagged phospho-mimetic MAP6 demonstrating restricted localization to actin-rich regions.

5. Is phosphorylation of MAP6 also required for the actin-based functions of the protein, such as the actin straightening and bundling?

6. On page 6, it is stated that both mushroom-shaped and stubby spines responded to KO of MAP6, however this effect was not fully born out in transfected hippocampal neurons. Please include a reason that mushroom but not stubby spines are more strikingly diminished by changes in MAP6 expression.

7. Please add to the body of the paper the explanation for selecting the R5 repeat of Mc in Figure S1. The rationale is explained in the figure legend, but would benefit from also being in the main body of the manuscript.

8. Please quantify the significance for Figure 4D on the far right panel, as it appears the baseline is already lowered and it is unclear if there is still a significant change with the induction of cLTP.

9. The discussion includes a paragraph on LIM kinase that hinges on an unpublished observation that LIM kinase can phosphorylate MAP6. Without including these data in the current manuscript so that they can be assessed directly, the connection is speculative.

10. An additional summary paragraph at the end of the manuscript's discussion would be helpful.

11. If you are going to mention He mice in the discussion, please state what mutation these mice have, as they were not previously mentioned.

Minor revisions:

- First paragraph in results "displayed significantly less dendritic spines" should read "fewer" not "less"

- Fig S1B, the KO+R5 image is blurry

- Fig S1G the purple bar needs significance indicated

- Fig 3 in the figure legend C' is listed as C" (so there are two C" in the legend)

- Fig 4D far right please show that there is a significant difference

- Fig 5B reverse the colors of the image so the actin is visible (black on white, rather than white on black)

- In the discussion, "Full-size MAP6 as well as the isolated Mc domains [[are]] concentrated in actin-rich regions of neurons" is missing the bracketed 'are'. This particular sentence also need to recognize that the staining is pan-neuronal and localizes with MTs as well as actin.

Reviewer #2 (Remarks to the Author):

This paper describes new results on the cytoskeletal associated protein MAP6, implicating it in the regulation of dendritic spines via direct actin filament binding. MAP6 (also known as STOP, for stable-tubule only protein) is a very interesting molecule, and this paper contributes to the accumulating evidence that its main functions in vivo do not pertain exclusively (or perhaps at all) to microtubule dynamics (as originally-assumed when it was first discovered). The strength of the paper is that it uses both knockout mice and cultured primary neurons to establish the influence of MAP6 on spines, and that it includes a series of in vitro biochemical studies using purified proteins to test whether and how MAP6 interacts directly with F-actin. Some of the results are quite interesting, but are not presented with sufficient context for a non-specialist reader to appreciate their significance. One of the most intriguing findings is the chem-LTP-induced decrease in spine F-actin photobleaching recovery, which is abolished in the MAP6 knockout mouse cultures. This finding deserves enhanced rigor in execution and interpretation, as well as some strong follow up experiments. The weakness of the paper is that none of the major concepts are new, so the present work makes a fairly incremental advance in the broader topic area, although one with potential impact in the MAP6 field. The idea that many cytoskeletal-associated proteins can bind both microtubules and actin filaments is by now well established. The authors' own previous work reported that MAP6 can co-localize in situ with subcellular compartments other than microtubules, and that MAP6 can colocalize with actin-rich compartments in neurons and can bind to F-actin in vitro. It would be appropriate for the authors to clarify in their paper what were the limitations of the previous work and how the present studies extend our knowledge on MAP6-actin interaction. By now there is a fairly long list of actin binding proteins that regulate dendritic spine development or stability, so this is not conceptually novel, although it is certainly interesting that MAP6 is a new member of the list.

Major comments/suggestions for improving rigor

1. FRAP experiments. These assays report one of the most interesting mechanistic aspects of the paper, yet they are poorly described. The Methods section needs a much more detailed description of the experimental design and interpretation, including the duration of the bleaching procedure, what was the criteria for complete bleaching, how were corrections applied for photobleaching during image collection, how were corrections for changes in spine volume applied, etc. The results section merely states that the "mobile fraction" of GFP-actin was calculated from "recovery kinetics." Authors need to describe their procedure for defining the mobile fraction (does this include monomeric actin, if not, why not?), what equation was used for the curve fitting function, and what underlying assumptions were made regarding the kinetics (one fit model, two fit model, and why)? Was the first-order rate constant calculated, and if not how can one define the fraction of "mobile" vs "stable" actin? Do the curves shown in Fig 4 reflect mean and standard error? If so, the data appear

surprisingly reproducible for the relatively low "n" (~20 spines per condition), compared to other reports in the literature. Were all shapes, categories (stubby, thin, mushroom) and sizes of spines included in the analysis, and if not, what selection criteria were used? The authors should quantify whether the concentration of F-actin per spine changes. Finally, it is unclear whether the number of replicates contained in this data series is adequate to draw robust quantitative conclusions (although the cLTP effect does seem very large, and its impressive that it is utterly abolished in the KO mice). FRAP studies of actin dynamics in spines are notoriously fraught with variability and other limitations, and it would be desirable for the main findings to be replicated using another technique, like photoactivation. The authors are encouraged to refer to the comprehensive review on the above issues by Koskinen and Hotulainen (2014), Measuring F-actin properties in dendritic spines, *Frontiers in Neuroanatomy*, vol 8, article 74.

2. In vitro actin binding & bundling assays (Fig 5). These data strengthen the conclusion of the paper regarding a direct interaction between F-actin and the Mc region of MAP6. The authors need to include a control for specificity of this interaction. All the experiments only compare a condition in which F-actin is alone to one in which Mc is added. Results would be strengthened by including a control group using an irrelevant protein of similar charge and mass (perhaps a scrambled sequence?) as the Mc, to establish that the Mc effects are specific.

3. To strengthen the conclusion regarding a direct binding interaction between F-actin & Mc it would be good to carry out a saturation curve to determine Kd & Bmax.

4. Over-stating conclusions. In some places the quality of the paper (which is mostly strong) is undermined by over-interpretation of the results or hyperbole in conclusion statements:

a. The decrease in dendritic spine density in knockout animals is modest (20-30%), therefore, the conclusion stating that "MAP6 is required for the formation and maintenance of mature post-synaptic spines" (p. 4), that "MAP6 plays a crucial role in the formation and/or maintenance of mature dendritic spines" (p. 6), or that there is a "critical function" for this protein in spine stabilization (title), appear to be over-statements. Clearly spines can form and even persist in the absence of MAP6, so although it may be important, it is not "critical."

b. (p. 10) "Like endogenous MAP6, the GFP-Mc construct was recruited by actin filaments in growth cones...." Colocalization to a subcellular compartment containing F-actin does not necessarily mean that the molecule is "recruited" by F-actin. (Also authors should not use the word 'construct' here... it's the protein, not the cDNA that is localized.)

c. (p. 10). "Overall these results clearly show a specific localization of endogenous MAP6...." This is an incorrect statement that over-interprets the otherwise very nice data. The results show that endogenous MAP6 appears to be present very extensively throughout the neuron (Fig. 1B), including (but not "specifically") at actin-rich structures, where (as indicated by the acceptor photobleaching experiment) it can be in close proximity to F-actin. If the authors wish to conclude that the localization is specific to F-actin, then they need to address why the immunostaining appears throughout the neuron, and why they do not detect it in Golgi, mitochondria, plasma membrane, etc, as they previously reported for

non-neuronal cells and hippocampal neurons (Gory-Faure et al 2014).

Additional comments/suggestions:

5. Fig. 1. For helping guide the reader through all the data, I suggest labeling the graphs to indicate whether they are from in vivo data or from cultured neurons.

6. Since the authors emphasize the fact that specific categories of spines are affected (and others not), it would be helpful for the reader to discuss more clearly the biological meaning of this observation (i.e., what it implies about mechanism and what is the functional impact... what are stubby spines good for, for example?).

7. Fig. 2E. The example shown for KO + MAP6-E delta Mc is not representative of the magnitude of the spine decrease, which is more modest than the example shown. The complete absence of spines in this example suggests it might be another class of neuron, i.e., not a spiny pyramidal type neuron, but an aspiny interneuron. Same goes for Fig S1F.

8. Fig. 3D-E. The acceptor photobleaching results are very nice. I suggest showing images for the GFP control, as this important experiment establishes specificity.

9. Fig. 3A-C. In the examples shown, the distribution of MAP6 or GFP-Mc look ubiquitous, and not specific to F-actin. Need to compare to a cell filler.

10. Fig. 4.A. It is difficult for the reader to locate the boxed region in the dendrite images. Would be good to include a box in the whole neuron images also.

Reviewers' comments:

Reviewer #1 (Remarks to the Author):

Peris et al demonstrate that the canonically microtubule-associated protein MAP6 co-localizes with actin in dendritic spines and the growth cone, and is there responsible for promoting nucleation and bundling of actin filaments. They report that MAP6 supports the formation or maintenance and the maturation of dendritic spines. The authors show that this effect is mediated by the Mc domain of MAP6, and relies on the CaMKII-dependent phosphorylation of MAP6 for disassociation from microtubules and subsequent actin relocalization.

Overall, the manuscript is well written, and the experiments are well thought out. The decision to focus briefly on LTP-associated changes using cLTP and the resulting figure are particularly striking. Additionally, the discussion was well-phrased and integrates previous observations in a compelling fashion, including the link to cofilin, septins, and non-MAP proteins known to cause striations in actin in other regions of the neuron. The detail in Figure 5H was stunning.

However, some clarifications and a few additional experiments would further increase the impact of the work, and minor revisions to the text would clarify the scope of this project.

We thanks the reviewers for the general comments and for the review of our work.

Specific points to address:

1. Experimentally, there is some concern about inconsistencies with the number of neurons examined for each condition, even within a single experiment. There are a number of times that 8-9 neurons are used for one experimental condition (generally KD), while the WT control and rescue conditions have double to triple the number of neuronal repeats. These experiments would be stronger if the groups being compared included similar numbers of observations. For example, Fig 4 there are only 10 neurons for the WT+cLTP when all other conditions have at least 20 neurons, and for 3E two conditions have 8 and 9 neurons, while some other conditions have up to 30 neurons, over three times the lowest value.

Given the variability in the number of MAP6 knock-out (and wild-type) mice that can be generated in any given litter, and the added variability in transfection efficiency, it is very difficult to obtain an equal number of assayed neurons for both genotypes. However we think that this does not affect the validity of our statistical calculations. While we agree that ideally samples should be balanced, unequal sample size would be a concern in the case of ANOVA with two (or more) factors, or if sample variance were significantly heterogeneous. Here we used a one-way ANOVA procedure and post hoc t tests that include a classical (Welch) correction for unequal sample sizes (the within-group variance being weighted group-wise by the number of degrees of freedom), as per the standard routine provided in the statistics software. We also checked the homogeneity of variance across groups (using the non-parametric Fligner test).

In Fig. 4, the cLTP effect in WT neurons is a confirmation of an observation that has been well documented in the literature, and even with $n=10$ neurons the change was highly significant in our data ($p=0.0002$, post hoc pairwise t test with correction for multiple comparisons). To compare our different conditions, in our manuscript we used again one-way ANOVA (considering a single "treatment" factor) for Fig. 4, thus averting errors due to the imbalance of the samples. We also checked the separate significance of the cLTP stimulation, KO genotype and stimulation x genotype

interaction effects, using two-way ANOVA with Type III sum of squares to control for sample imbalance.

For Fig. 3E, the n values correspond to growth cones of differentiating neurons (DIV2), not to neurons. GFP is a negative control inducing negligible FRET. The probability of no significant FRET following rescue with MAP6 or Mc is vanishingly small ($p < 2.10^{-16}$). In case there was a typo and the reviewer was in fact referring to Fig. 4E, even with discrepancies between n values the results are highly significant, for the same reasons as explained above.

To simplify the reading of figure legends, we now indicate in the text the size (n) of the smallest sample in each series; the list of all n values is provided in supplemental data.

2. The authors quantify spine density in many figures, but there seems to be a striking change in spine area as well. Has this been examined?

Using NeuronStudio, we have now measured both spine head diameter and head volume specifically in *mushroom* spines and found no significant difference in diameter between WT and KO neurons.

head volume			head diameter		
	WT	KO		WT	KO
Number of values	11	8	Number of values	11	8
Mean	0.3664	0.2941	Mean	0.4571	0.4464
Std. Deviation	0.1309	0.07964	Std. Deviation	0.02917	0.06421
Std. Error of Mean	0.03947	0.02816	Std. Error of Mean	0.008795	0.0227
Unpaired t test			Unpaired t test		
P value	0.1848		P value	0.6315	
P value summary	ns		P value summary	ns	

As no difference in spine volume or diameter was found between WT and KO, we did not include these new data in the revised manuscript, in order to keep the emphasis on the difference in **spine density** between WT and KO neurons.

3. The largest problem is the inconsistency with which MAP6's interaction with microtubules is explained. Multiple times a separate paper is referenced which posits that MAP6 only binds microtubules at lower temperatures, yet throughout this manuscript MAP6 is seen co-localizing with microtubules. Within the manuscript it is both stated that "as Mc domains cannot bind MTs at physiologic temperature" (page 9) but also that "endogenous MAP6 co-localized with MTs" (page 10). Please demonstrate that other domains are responsible for MAP6 binding to MTs or find a different way to explain these divergent findings.

We apologize for our confused statements about MAP6 interaction with microtubules.

MAP6 binds to microtubules through 2 types of microtubule-binding modules, i.e. the **Mn domains** and the **Mc domains** depicted in the figure below (Fig. 2A; Bosc et al, 2001).

Mn domains (in orange here) interact with microtubules both at physiological temperature and at 4° C, whereas Mc domains (in grey here) interact with microtubules only at 4°C (Delphin et al, 2012). Thus only Mn domains are responsible for microtubule binding at physiological temperature.

We changed the text page 10 to clarify these points.

4. Throughout the manuscript, MAP6 is stated as localizing with actin in particular, but this overlooks the obvious co-localization within the dendrite and the axon with MTs. MAP6 expression appears to be pan-neuronal. It would help greatly if staining was able to be conducted for phospho-MAP6 to demonstrate spine and growth cone specificity, or if KD neurons had forced expression of tagged phospho-mimetic MAP6 demonstrating restricted localization to actin-rich regions.

Concerning "phospho-MAP6 staining" : We agree with the reviewer that phospho-MAP6 labeling would have been of interest. Accordingly we had found preferential association of phospho-MAP6 with actin (both in dendritic spines, growth cone, axonal branching point) as published in Baratier et al, 2006.

The corresponding figure of Baratier's paper is shown below (MAP6 is indicated as STOP and phospho MAP6 as STOP-P).

[Redacted: Fig. 5b from Baratier J, *et al.* Phosphorylation of microtubule-associated protein STOP by calmodulin kinase II. *J Biol Chem* 281, 19561-19569 (2006).]

Unfortunately, our anti-phospho MAP6 antibody is no more functional, precluding new labeling.

Concerning "forced expression of tagged phospho-mimetic MAP6 and ... restricted localization to actin-rich regions" : phospho-mimetic MAP6 expressed in WT or MAP6KO was not restricted to the spine. This is very likely due to over-expression, as other over-expressed proteins including isolated Mc domains also existed in the dendritic shaft, despite the fact that Mc domains are not retained on

dendritic microtubules [GFP-Mc (Fig 2E), GFP-R4-R5, GFP-R5, human Mc-GFP (Fig S1 B-D) and actin-GFP (Fig 4A)].

The point we wanted to make was that MAP6, on top of its ability to interact with MTs, has the ability to interact with actin in some conditions, we modify the text and add a paragraph at the end of the sub section related to actin localization of MAP6 (p11) to better explain these points.

5. Is phosphorylation of MAP6 also required for the actin-based functions of the protein, such as the actin straightening and bundling?

Regarding the effect of phosphorylation by CamKII, we have previously shown that CAMKII phosphorylated forms of MAP6 no longer interacted with microtubules, but did bind to actin. We proposed that CAMKII phosphorylation of MAP6 allowed its detachment from microtubules and its interaction with actin cytoskeleton, which we now show occurs via the Mc domain. However, CamKII phosphorylation takes place outside of the Mc domains. Whether these latter contain phosphorylation sites recognized by some other kinase is currently not known. A sentence was added in the discussion (p 20) to stress this point.

We agree with the reviewer that the possible role of phosphorylation in actin bundling or straightening is an important question, but it falls beyond the scope of the present paper.

6. On page 6, it is stated that both mushroom-shaped and stubby spines responded to KO of MAP6, however this effect was not fully born out in transfected hippocampal neurons. Please include a reason that mushroom but not stubby spines are more strikingly diminished by changes in MAP6 expression.

We thank the reviewer for pointing out this discrepancy. As now mentioned in the text (p6), the difference might be attributable to the details of spine maturation, which obviously differ between *in vivo* and *in vitro* conditions. Difficulties in correctly differentiating stubby spines from mushroom spines with a very short neck by optical microscopy probably compounds the problem. However, we note that acute KD of endogenous MAP6 in WT neurons also resulted in a drop in stubby spines. Importantly, in all experimental systems we used (*in vivo*, *in vitro* cortical and/or hippocampal neurons, KO and or siRNA) the knockdown of MAP6 affected the density of mushroom like spines. We focus on this finding in the text (p6).

7. Please add to the body of the paper the explanation for selecting the R5 repeat of Mc in Figure S1. The rationale is explained in the figure legend, but would benefit from also being in the main body of the manuscript.

The text has been modified and we now present (in the new Fig. S1B) a sequence alignment of the 5 Mc modules found in rat MAP6, against the single Mc module of the human protein.

8. Please quantify the significance for Figure 4D on the far right panel, as it appears the baseline is already lowered and it is unclear if there is still a significant change with the induction of cLTP.

The significance of the change induced by cLTP was indeed established, based not on average FRAP curves only shown for illustration (new Fig. 4F, right panel), but on kinetic parameters calculated by non-linear regression of each individual curve onto a mono-exponential recovery model, as now described in detail in the Materials and Methods section, p.28. The new Fig. 4H, right panel, shows the significant difference between mobile fractions under control and cLTP conditions.

9. The discussion includes a paragraph on LIM kinase that hinges on an unpublished observation that LIM kinase can phosphorylate MAP6. Without including these data in the current manuscript so that they can be assessed directly, the connection is speculative.

The paragraph has been removed.

10. An additional summary paragraph at the end of the manuscript's discussion would be helpful.

We now added a summary paragraph at the end of the discussion.

11. If you are going to mention He mice in the discussion, please state what mutation these mice have, as they were not previously mentioned.

We apologize for using the abbreviation "He" instead of the more explicit "heterozygous" designation. We changed this in the manuscript.

Minor revisions:

- First paragraph in results "displayed significantly less dendritic spines" should read "fewer" not "less" We corrected the manuscript.

- Fig S1B, the KO+R5 image is blurry We have corrected image of KO+R5.

- Fig S1G the purple bar needs significance indicated We added "ns" to indicate non-significant differences.

- Fig 3 in the figure legend C' is listed as C" (so there are two C" in the legend) We changed figure legend.

- Fig 4D far right please show that there is a significant difference. Please see explanation above.

- Fig 5B reverse the colors of the image so the actin is visible (black on white, rather than white on black) We changed images in the figure.

- In the discussion, "Full-size MAP6 as well as the isolated Mc domains [[are]] concentrated in actin-rich regions of neurons" is missing the bracketed 'are'. This particular sentence also need to recognize that the staining is pan-neuronal and localizes with MTs as well as actin. We modified the sentence in the discussion.

Reviewer #2 (Remarks to the Author):

This paper describes new results on the cytoskeletal associated protein MAP6, implicating it in the regulation of dendritic spines via direct actin filament binding. MAP6 (also known as STOP, for stable-tubule only protein) is a very interesting molecule, and this paper contributes to the accumulating evidence that its main functions in vivo do not pertain exclusively (or perhaps at all) to microtubule dynamics (as originally-assumed when it was first discovered). The strength of the paper is that it uses both knockout mice and cultured primary neurons to establish the influence of MAP6 on spines, and that it includes a series of in vitro biochemical studies using purified proteins to test whether and how MAP6 interacts directly with F-actin. Some of the results are quite interesting, but are not presented with sufficient context for a non-specialist reader to appreciate their significance. One of the most intriguing findings is the chem-LTP-induced decrease in spine F-actin photobleaching recovery, which is abolished in the MAP6 knockout mouse cultures. This finding deserves enhanced rigor in execution and interpretation, as well as some strong follow up experiments. The weakness of the paper is that none of the major concepts are new, so the present work makes a fairly incremental advance in the broader topic area, although one with potential impact in the MAP6 field. The idea that many cytoskeletal-associated proteins can bind both microtubules and actin filaments is by now well established. The authors' own previous work reported that MAP6 can co-localize in situ with subcellular compartments other than microtubules, and that MAP6 can colocalize with actin-rich compartments in neurons and can bind to F-actin in vitro. It would be appropriate for the authors to clarify in their paper what were the limitations of the previous work and how the present studies extend our knowledge on MAP6-actin interaction. By now there is a fairly long list of actin binding proteins that regulate dendritic spine development or stability, so this is not conceptually novel, although it is certainly interesting that MAP6 is a new member of the list.

We thanks the reviewers for the general comments and for the review of our work.

Major comments/suggestions for improving rigor

1. FRAP experiments. These assays report one of the most interesting mechanistic aspects of the paper, yet they are poorly described. The Methods section needs a much more detailed description of the experimental design and interpretation, including the duration of the bleaching procedure, what was the criteria for complete bleaching, how were corrections applied for photobleaching during image collection, how were corrections for changes in spine volume applied, etc. The results section merely states that the "mobile fraction" of GFP-actin was calculated from "recovery kinetics." Authors need to describe their procedure for defining the mobile fraction (does this include monomeric actin, if not, why not?), what equation was used for the curve fitting function, and what underlying assumptions were made regarding the kinetics (one fit model, two fit model, and why)? Was the first-order rate constant calculated, and if not how can one define the fraction of "mobile" vs "stable" actin? Do the curves shown in Fig 4 reflect mean and standard error? If so, the data appear surprisingly reproducible for the relatively low "n" (~20 spines per condition), compared to other reports in the literature. Were all shapes, categories (stubby, thin, mushroom) and sizes of spines included in the analysis, and if not, what selection criteria were used? The authors should quantify whether the concentration of F-actin per spine changes. Finally, it is unclear whether the number of replicates contained in this data series is adequate to draw robust quantitative conclusions (although the cLTP effect does seem very large, and its impressive that it is utterly abolished in the KO mice).

FRAP studies of actin dynamics in spines are notoriously fraught with variability and other limitations, and it would be desirable for the main findings to be replicated using another technique, like photoactivation. The authors are encouraged to refer to the comprehensive review on the above issues by Koskinen and Hotulainen (2014), Measuring F-actin properties in dendritic spines, *Frontiers in Neuroanatomy*, vol 8, article 74.

The Methods section needs a much more detailed description of the experimental design and interpretation, including the duration of the bleaching procedure, what was the criteria for complete bleaching, how were corrections applied for photobleaching during image collection, how were corrections for changes in spine volume applied, etc.

We deeply apologize for the insufficient description of our FRAP methods. We have actually followed standard guidelines for performing and analyzing these experiments. The procedure is now reported in detail in the Materials and Methods section.

The results section merely states that the “mobile fraction” of GFP-actin was calculated from “recovery kinetics.” Authors need to describe their procedure for defining the mobile fraction (does this include monomeric actin, if not, why not?), what equation was used for the curve fitting function, and what underlying assumptions were made regarding the kinetics (one fit model, two fit model, and why)? Was the first-order rate constant calculated, and if not how can one define the fraction of “mobile” vs “stable” actin? Do the curves shown in Fig 4 reflect mean and standard error? If so, the data appear surprisingly reproducible for the relatively low “n” (~20 spines per condition), compared to other reports in the literature.

As now explained in the Methods section, the mobile fraction was defined as factor A in the equation $[y = y_0 - A \cdot \exp(-k \cdot x)]$ corresponding to a mono-exponential recovery model. The mobile fraction as well as the other parameters (including the first-order rate constant k) were calculated for each individual FRAP assay, by non-linear regression to the model. Goodness of fit was not improved when using a two-phase recovery model, therefore we retained the simpler monophasic curve. The mean and standard error of mobile fraction values (shown in the new Figs. 4F, 4H) were then calculated from groups of individual fits, corresponding to each genotype / stimulation condition. The curves (new Figs. 4E, 4G) were obtained by calculating the mean and standard error at each time point, and are shown only for illustration purposes. Diffusional entry of monomeric actin was not included in the model because in our hands it occurs with a time scale that is at least ≈ 40 -fold faster than polymerization; any effect of diffusion is thus included in the value of the first recovery time point.

Were all shapes, categories (stubby, thin, mushroom) and sizes of spines included in the analysis, and if not, what selection criteria were used?

For FRAP experiments, we selected mushroom-like spines since these are more stable, appropriate for accurate measurements, and relatively homogeneous in terms of responsiveness to synaptic activation. It is now indicated in the text p11.

The authors should quantify whether the concentration of F-actin per spine changes.

Changes in spine volume and F-actin concentration have been measured and are now shown in the new Figs. 4B and 4C. The results regarding WT neurons are in agreement with published data and those for KO neurons are completely in line with the other experiments, indicating a profound deficiency in actin cytoskeleton expansion.

Finally, it is unclear whether the number of replicates contained in this data series is adequate to draw robust quantitative conclusions (although the cLTP effect does seem very large, and its impressive that it is utterly abolished in the KO mice). FRAP studies of actin dynamics in spines are notoriously fraught with variability and other limitations, and it would be desirable for the main findings to be replicated using another technique, like photoactivation. The authors are encouraged to refer to the comprehensive review on the above issues by Koskinen and Hotulainen (2014), Measuring F-actin properties in dendritic spines, Frontiers in Neuroanatomy, vol 8, article 74.

As reckoned by the reviewer, the size of the cLTP effect and that of the KO effect are surprisingly large, so that the number of replicates in our data series is indeed sufficient for differences to reach quite a low p value in the WT condition under the null hypothesis, and also quite a high power in the KO condition under the alternative hypothesis. Since we took care to check the usual assumptions (normality, variance homogeneity) for our statistical calculations, as now described in Materials and Methods, we believe that the results are statistically valid by generally recognized criteria. The effect size was also such that following the recommendations in Koskinen and Hotulainen (2014), in our case the FRAP technique was suitable to assay changes in actin dynamics. Therefore we think that while interesting in itself, studying actin dynamics by photo-activation goes beyond the scope of the present paper and will not strongly improve the general outcome.

We now provide detailed technical description of the acquisition procedure as well as of data treatment.

2. In vitro actin binding & bundling assays (Fig 5). These data strengthen the conclusion of the paper regarding a direct interaction between F-actin and the Mc region of MAP6. The authors need to include a control for specificity of this interaction. All the experiments only compare a condition in which F-actin is alone to one in which Mc is added. Results would be strengthened by including a control group using an irrelevant protein of similar charge and mass (perhaps a scrambled sequence?) as the Mc, to establish that the Mc effects are specific.

3. To strengthen the conclusion regarding a direct binding interaction between F-actin & Mc it would be good to carry out a saturation curve to determine K_d & B_{max} .

In accordance with the reviewer's suggestion, the specificity of the Mc-actin interaction is now demonstrated by the saturation isotherm that we provide. The new Fig. 5A,B shows the data from high-speed co-sedimentation assays and the resulting saturation curve, from which values have been calculated for K_d and B_{max} . The K_d value (about 670 nM) indicates significant binding affinity. Please note that in the large number of actin-protein interactions that have been analyzed in the literature, and especially in previous work by one of our teams (L. Blanchoin), effects on actin filament organization similar to those of Mc modules have never been observed. We believe that this fact also strongly argues for the specificity of the Mc-actin interaction, making it less critical to repeat the work with irrelevant proteins as controls.

4. Over-stating conclusions. In some places the quality of the paper (which is mostly strong) is undermined by over-interpretation of the results or hyperbole in conclusion statements:

a. The decrease in dendritic spine density in knockout animals is modest (20-30%), therefore, the conclusion stating that "MAP6 is required for the formation and maintenance of mature post-synaptic spines" (p. 4), that "MAP6 plays a crucial role in the formation and/or maintenance of mature dendritic spines" (p. 6), or that there is a "critical function" for this protein in spine

stabilization (title), appear to be over-statements. Clearly spines can form and even persist in the absence of MAP6, so although it may be important, it is not “critical.”

We agree with the reviewer's comments concerning the not critical role of MAP6 in "formation and maintenance of mature spines... etc" and we toned down our statements in the text (p4 and p6). We also replaced "critical" in the title with "key".

b. (p. 10) “Like endogenous MAP6, the GFP-Mc construct was recruited by actin filaments in growth cones...” Colocalization to a subcellular compartment containing F-actin does not necessarily mean that the molecule is “recruited” by F-actin. (Also authors should not use the word ‘construct’ here... it’s the protein, not the cDNA that is localized.)

This part of the text has been modified p10 and please also see reply to question 9.

c. (p. 10). “Overall these results clearly show a specific localization of endogenous MAP6...” This is an incorrect statement that over-interprets the otherwise very nice data. The results show that endogenous MAP6 appears to be present very extensively throughout the neuron (Fig. 1B), including (but not “specifically”) at actin-rich structures, where (as indicated by the acceptor photobleaching experiment) it can be in close proximity to F-actin. If the authors wish to conclude that the localization is specific to F-actin, then they need to address why the immunostaining appears throughout the neuron, and why they do not detect it in Golgi, mitochondria, plasma membrane, etc, as they previously reported for non-neuronal cells and hippocampal neurons (Gory-Faure et al 2014).

We agree with the reviewer's comment and modified the text accordingly p10 and 11. We also modify the text and add a paragraph at the end of the sub section related to actin localization of MAP6 (p11) to better explain these points.

Additional comments/suggestions:

5. Fig. 1. For helping guide the reader through all the data, I suggest labeling the graphs to indicate whether they are from in vivo data or from cultured neurons.

We thank the reviewer for his/her comment and we added this information in the Figure.

6. Since the authors emphasize the fact that specific categories of spines are affected (and others not), it would be helpful for the reader to discuss more clearly the biological meaning of this observation (i.e., what it implies about mechanism and what is the functional impact... what are stubby spines good for, for example?).

We add sentence in the discussion p18 to highlight the role of Mc modules in actin stabilization in matures/active spines.

7. Fig. 2E. The example shown for KO + MAP6-E delta Mc is not representative of the magnitude of the spine decrease, which is more modest than the example shown. The complete absence of spines in this example suggests it might be another class of neuron, i.e., not a spiny pyramidal type neuron, but an aspiny interneuron. Same goes for Fig S1F.

We apologize again for the lack of clarity of our text which has led to a misunderstanding. In Figure 2E and S1F, mCherry was used as a cell filler while GFP labels the different proteins used for rescue. In Fig. 2E, spines are readily apparent in the mCherry channel, albeit with a reduced density as described elsewhere in the paper. The lack of label in spines in Figs. 2E and S1F (green channel)

reveals that the GFP fusion protein fails to enter spines that are otherwise visible in the red channel. We modified the text in the results section p7 to indicate this more clearly.

8. Fig. 3D-E. The acceptor photobleaching results are very nice. I suggest showing images for the GFP control, as this important experiment establishes specificity.

We now added images for the GFP Control. To clarify our apFRET experiments, a movie of a pre- and post-acceptor bleached images of GFP or MAP6-E GFP expressing neurons is now provided in Supplementary Movie1. Acceptor bleached regions are indicated with an arrowhead. As observed with the large increase of MAP6-E GFP (donor) fluorescence, Förster energy transfer between MAP6 and actin is evident in growth cones when the acceptor (TRITC Phalloidin) is bleached.

9. Fig. 3A-C. In the examples shown, the distribution of MAP6 or GFP-Mc look ubiquitous, and not specific to F-actin. Need to compare to a cell filler.

We agree with the reviewer that endogenous MAP6 or GFP-Mc distribution are not exclusive to F-actin. We modified the text p10 and 11. In addition, to compare the distribution of GFP-Mc with that of unfused GFP, we used MEFs because they are flat cells in which specific overlap with actin filaments is easier to observe than in neurons. The new Fig. S2 shows that a pool of GFP-Mc is localized along actin stress fibers (panel B), whereas unfused GFP remains completely diffuse (panel A).

10. Fig. 4.A. It is difficult for the reader to locate the boxed region in the dendrite images. Would be good to include a box in the whole neuron images also.

Fig. 4 has been modified. The high magnification images have been improved and the image of whole neurons have been removed, as suggested by Reviewer 1.

REVIEWERS' COMMENTS:

Reviewer #1 (Remarks to the Author):

The authors have responded to the points raised in the previous round of review in a thorough and thoughtful manner. Their responses are generally satisfactory, with the exception of the response to the first point. They respond to the concern about the unbalanced data set with a statistical analysis, which is fine but then indicate that they will make this point opaque to most readers by only listing the smallest sample size in each series within the figure legend. While they propose to list the n values for all data sets in the supplement, only the most tenacious of readers will take the time to reason all of this out. In order to be most transparent, given the wide range of n values used in the studies reported, this point should be clear within the main text allowing readers to draw their own conclusions.

Reviewer #2 (Remarks to the Author):

The revised manuscript does a more thorough job of explaining methodology and clarifying some confusing points regarding the nature of the microtubule and actin interactions for MAP6. They have also generally toned down their over-statements regarding the role for MAP6 in spines, and mostly removed claims that MAP6 is "concentrated" in actin-rich areas, although on (line 399-400) of the Discussion there remains a sentence that reads "Mc domain are concentrated in actin-rich regions of neurons, such as growth cones of young neurons or dendritic spines in mature cells." This reviewer suggests replacing the word "concentrated" with the word "present," because the data do not support the idea that MAP6 is particularly enriched in these areas relative to many others (a point made by both referees).

It is disappointing that the authors made only a little effort to add experimental data to the paper to address concerns raised by both referees regarding rigor and reproducibility. It is nice that the authors now included a curve to demonstrate that Mc binding to F-actin in vitro is saturable (however, these data do not establish binding specificity, as they mistakenly stated in their rebuttal letter). They still chose not to include a control for specificity for the in vitro experiments. Instead, the rebuttal letter argues that "effects on actin filament organization similar to those of Mc modules have never been observed." This seems a fairly weak argument on whether charge and sequence specificity are important for binding, and for the apparent effects of the Mc on actin filament organization. At minimum, the Discussion should acknowledge this limitation and present an appropriately referenced argument for MAP6 exhibiting unique features relative to other ABPs. The readers can then judge for themselves. Finally, it is a shame that the authors added nothing to strengthen their interesting but fairly preliminary observations on the effects MAP6 on actin filament turnover following LTP-like stimulation, which contains the conceptually newest observations regarding this protein. The new description in Methods establishes that these experiments were performed rigorously using accepted methodology and are statistically robust.

In summary, the manuscript is modestly strengthened from its previous version, and the data support the authors' major conclusions.

REVIEWERS' COMMENTS:

Reviewer #1 (Remarks to the Author):

The authors have responded to the points raised in the previous round of review in a thorough and thoughtful manner. Their responses are generally satisfactory, with the exception of the response to the first point. They respond to the concern about the unbalanced data set with a statistical analysis, which is fine but then indicate that they will make this point opaque to most readers by only listing the smallest sample size in each series within the figure legend. While they propose to list the n values for all data sets in the supplement, only the most tenacious of readers will take the time to reason all of this out. In order to be most transparent, given the wide range of n values used in the studies reported, this point should be clear within the main text allowing readers to draw their own conclusions.

We add back the exact n in all figures legends (in green in the text).

Reviewer #2 (Remarks to the Author):

The revised manuscript does a more thorough job of explaining methodology and clarifying some confusing points regarding the nature of the microtubule and actin interactions for MAP6. They have also generally toned down their over-statements regarding the role for MAP6 in spines, and mostly removed claims that MAP6 is “concentrated” in actin-rich areas, although on (line 399-400) of the Discussion there remains a sentence that reads “Mc domain are concentrated in actin-rich regions of neurons, such as growth cones of young neurons or dendritic spines in mature cells.” This reviewer suggests replacing the word “concentrated” with the word “present,” because the data do not support the idea that MAP6 is particularly enriched in these areas relative to many others (a point made by both referees).

We are sorry for forgetting this change. We modified "concentrated " to "present"

It is disappointing that the authors made only a little effort to add experimental data to the paper to address concerns raised by both referees regarding rigor and reproducibility. It is nice that the authors now included a curve to demonstrate that Mc binding to F-actin in vitro is saturable (however, these data do not establish binding specificity, as they mistakenly stated in their rebuttal letter). They still chose not to include a control for specificity for the in vitro experiments. Instead, the rebuttal letter argues that “effects on actin filament organization similar to those of Mc modules have never been observed.” This seems a fairly weak argument on whether charge and sequence specificity are important for binding, and for the apparent effects of the Mc on actin filament organization.

As compared to the original version the revised version encloses a saturation curve of Mc on actin filaments. We found that actin filaments have binding sites of strong affinity for Mc, i.e. below the μM range. In the same biochemical conditions, well-characterized actin filament bundling proteins such as villin have a similar affinity ($K_d=670$ nM) for actin filaments (Huang et al., Plant Cell, 2005). We thus believe we have answered the reviewer's concern .

At minimum, the Discussion should acknowledge this limitation and present an appropriately referenced argument for MAP6 exhibiting unique features relative to other ABPs. The readers can then judge for themselves.

The discussion has been revised and now states the limitations of our study in a more explicit way, e.g. we caution that the effects of MAP6 Mc on actin filament organization have only been observed in vitro. We also discuss in more details the effects of a range of other ABPs on actin organization as compared to Mc modules, allowing the reader to evaluate the specificity of the Mc effect.

Finally, it is a shame that the authors added nothing to strengthen their interesting but fairly preliminary observations on the effects MAP6 on actin filament turnover following LTP-like stimulation, which contains the conceptually newest observations regarding this protein.

As said before, we indeed think that the effect size of MAP6 Mc modules on actin in spines was such that following the recommendations in Koskinen and Hotulainen (2014), in our case the FRAP technique was suitable to assay changes in actin dynamics. Therefore we still think that while interesting in itself, studying actin dynamics by photo-activation will not strongly improve the general outcome.

The new description in Methods establishes that these experiments were performed rigorously using accepted methodology and are statistically robust.

In summary, the manuscript is modestly strengthened from its previous version, and the data support the authors' major conclusions.